# Doppler Prompting for Stable mmWave-Based Human Pose Estimation

Shuntian Zheng [1]   Jiaqi Li [1]   Xiaoman Lu [1]   Shuai He [2]   Yu Guan* [1]

## Abstract

Millimeter-wave (mmWave) enables privacy-preserving and illumination-robust human pose estimation (HPE), with each mmWave frame represented as a range–angle–Doppler tensor, providing spatial magnitude for localization and Doppler signatures for motion-related cues. However, existing mmWave-based HPE methods either underutilize or naïvely fuse Doppler signatures with spatial magnitude, disregarding their distinct physical semantics. As a result, non-human Doppler signatures can be misinterpreted as human motion cues, leading to jittery trajectories. We propose **PULSE**, which converts Doppler signatures into confidence-aware motion prompts and injects them into spatial magnitude reasoning through constrained interactions. By screening Doppler prompts before they influence prediction, PULSE first suppresses spurious spectral motion cues and then uses the screened prompts to stabilize prediction. Across three datasets spanning single- and multi-person settings, PULSE consistently improves pose accuracy and temporal stability, indicating that controlled Doppler prompting is a practical direction for stable mmWave HPE. Codes are available in here.

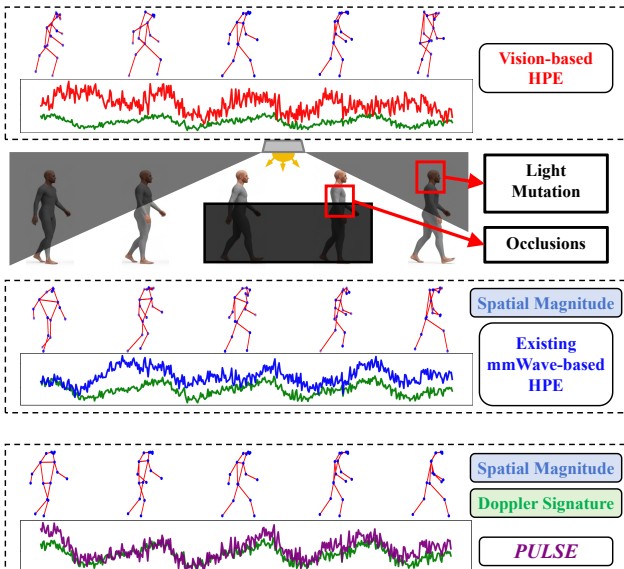

*Figure 1.* HPE under disturbed inference. Top: Vision-based systems can degrade under illumination shifts and occlusions. Middle: Existing mmWave HPE methods can exhibit inter-frame fluctuations when evaluated over sequences. Bottom: PULSE yields more stable trajectories via controlled Doppler prompting.

## 1. Introduction

Accurate and **stable** human pose estimation (HPE) is essential in downstream applications that interpret predictions as trajectories rather than isolated frames, such as fall-risk assessment, rehabilitation tracking, and long-term monitoring (Li et al., 2022; Choi et al., 2023). In these settings, a pose sequence that is accurate at individual frames can still

be **operationally unusable** if it exhibits jittery motion or inconsistent short-term dynamics, since such artifacts can trigger false alarms or distort derived motion statistics.

However, obtaining stable trajectories is difficult when visual observations are intermittent or unreliable. In safety and health monitoring, continuous video capture and storage can be undesirable or infeasible due to privacy constraints (Choi et al., 2025), along with illumination changes and frequent occlusions as illustrated in Fig. 1.

Millimeter-wave (mmWave) provides a privacy-preserving sensing channel for HPE that is insensitive to lighting and visual occlusions because it measures electromagnetic reflections rather than appearance (Sengupta & Cao, 2022; Mei et al., 2024). This paper asks the question: **Can we obtain temporally stable HPE from mmWave under frame-wise inference without tracking or temporal smoothing?**

A mmWave frame is not just an image-like map: it is produced by Frequency-Modulated Continuous Wave (FMCW) processing, where the radar transmits a sequence of chirps

*Corresponding author [1]Department of Computer Science, University of Warwick, Coventry, United Kingdom [2]Department of Computer Science, Beijing University of Posts and Telecommunications, Beijing, China. Correspondence to: Yu Guan <Yu.Guan@warwick.ac.uk>.

*Proceedings of the 43rd International Conference on Machine Learning*, Seoul, South Korea. PMLR 306, 2026. Copyright 2026 by the author(s).

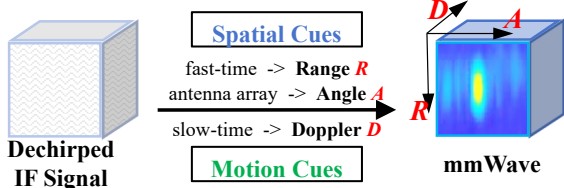

*Figure 2.* mmWave frame Fast Fourier Transform (FFT) formation. The dechirped IF signal is Fourier transformed along fast time to yield $R$; a slow-time Fourier transform across chirps yields $D$; and signals across antennas are combined to yield angle bins $A$.

and analyzes their returned signals (Richards et al., 2005). In plain terms, one can view each frame as data indexed by (1) *range* and *angle*, describing **where** strong reflectors appear, and (2) *Doppler spectrum*, describing **how** reflectors move along the radar's perceived direction within the frame based on Doppler principle (Chen et al., 2006). That is, after standard FFT-based processing as shown in Fig. 2, a mmWave frame can be represented as $\mathbf{H}_t \in \mathbb{R}^{R \times A \times D}$, where $R$ $A$ index spatial bins and $D$ indexes Doppler bins.

This dual nature suggests an appealing route to stability in HPE: range–angle responses support localization and body structure, while Doppler-derived motion provides instantaneous dynamics cues that can regularize short-term pose evolution. Crucially, this motion cue is already embedded **within a single frame** because Doppler is estimated from multiple chirps (a short temporal aperture) inside the frame (Richards et al., 2005); thus, stable frame-wise prediction is a sensible and practically relevant direction, rather than an ill-posed demand for motion from a static snapshot.

Designing a model that benefits from Doppler cues, however, is challenging. Doppler signatures are motion-sensitive but **not human-exclusive**: clutter, multipath reflections, and hardware-dependent effects can introduce spurious motion responses that resemble limb motion in the spectrum (Richards et al., 2005). If such nuisance responses are treated as equally reliable features and fused into pose reasoning, the model can mistake non-human spectral activity as motion evidence, which then propagates into frame-to-frame jitter and degraded temporal stability.

Current mmWave-based HPE methods generally exhibit three limitations regarding Doppler utilization. **First**, many methods predominantly rely on spatial magnitude and largely ignore Doppler cues, which limits their ability to constrain short-term dynamics under frame-wise evaluation (Zhu et al., 2024). **Second**, methods employing naïve integration (e.g., direct concatenation) overlook the heterogeneous reliability of Doppler signals caused by nuisance factors, inevitably introducing non-human motion noise into localization features (Choi et al., 2025). **Third**, another line of work improves stability by aggregating multiple frames, but it imposes strict test-time assumptions

regarding sequence continuity and still remains unable to effectively leverage Doppler cues when local spectra are unreliable (Kini et al., 2025).

We reconceptualize Doppler not as a symmetric feature channel, but as a **screened motion prompt** that conditionally guides spatial reasoning. Rather than competing with spatial features globally, Doppler serves as a reliability-gated cautious "hint", highlighting regions with high-confidence human motion evidence. In this paper, prompting denotes gated conditional-attention modulation of spatial reasoning by Doppler cues.

Based on this insight, we propose Prompting Using Local Spectral Estimates (**PULSE**), which comprises three components: (1) A **Magnitude Structural Stream** that extracts range–angle magnitude for robust localization and posture inference; (2) A **Motion Cue Stream** that refines Doppler signatures via reliability-aware weighting, suppressing spurious non-human signals prior to integration; (3) An **Alignment and Prompting** mechanism that maintains spatial correspondence and injects validated motion prompts exclusively within local neighborhoods to enforce stability.

Our contributions are summarized as follows:

(1) We analyze the role of Doppler in mmWave-based HPE and empirically observe that naïve fusion can yield limited temporal gains, highlighting a signal-level bottleneck that calls for reliability-aware exploitation of Doppler cues.

(2) We propose **PULSE**, which converts Doppler signatures into confidence-aware motion priors and selectively conditions spatial reasoning, offering a principled and plug-in interface for incorporating motion cues.

(3) Experiments across three datasets and single-/multi-person settings demonstrate PULSE's consistent improvements in per-frame accuracy and temporal stability; PULSE can also serve as a plug-in replacement, boosting performance across all datasets, substantiating controlled Doppler utilization as a practical design for stable mmWave HPE.

**Conflict of Interest Disclosure.** The authors declare that this work has no financial conflict of interest to disclose.

## 2. Background and Motivation

### 2.1. Preliminaries

mmWave radar operates by emitting frequency-modulated electromagnetic waves and measuring reflections. Using FMCW processing (Richards et al., 2005), one mmWave frame captures both spatial structure and motion of the scene. In the context of HPE, these features naturally decompose into two complementary domains with distinct functional roles, as shown in Fig. 3. Each frame can be modeled as $\mathbf{H} \in \mathbb{R}^{R \times A \times D}$, where $R$, $A$, and $D$ denote the range, angle,

and Doppler axis, and two domains are derived: spatial magnitude $(R, A)$ and Doppler signature $(D)$.

### 2.1.1. SPATIAL MAGNITUDE: RANGE AND ANGLE

Spatial magnitude describes the distribution of reflecting surfaces (Richards et al., 2005): The range dimension corresponds to the radial distance between the radar and the reflecting points and is obtained by analyzing FMCW chirps. Angles are recovered using antenna arrays, where phase differences across multiple transmit–receive elements encode the direction of arrival. In practice, modern radars estimate azimuth and elevation angles using Multiple-Input Multiple-Output (MIMO) or L-shaped arrays (Rahman et al., 2024). We denote the angular axis as $A$, which corresponds to azimuth that provides a single angle axis and extends naturally to azimuth–elevation grids when available. By combining these, each frame produces a spatial magnitude map on the $(R, A)$ plane. In HPE, $(R, A)$ provides geometric cues for body locations and structure (Zhu et al., 2024). However, spatial magnitude primarily reflects static or slowly varying scattering structure and is insensitive to instantaneous motion (Fan et al., 2024), limiting its ability to constrain inter-frame pose evolution.

### 2.1.2. DOPPLER SIGNATURE

Motion in mmWave is captured by slow-time phase progression and its corresponding Doppler spectrum (Richards et al., 2005). Relative motion along the radar line of sight introduces characteristic slow-time variations in the reflected signal, which are revealed by Fourier analysis along the slow-time dimension. As a result, each spatial location $(r, a)$ is associated with a Doppler spectrum that encodes the direction and magnitude of local motion.

In mmWave sensing, Doppler signatures provide motion-sensitive cues through slow-time spectral analysis (Thayaparan et al., 2007), but these cues are not exclusively determined by humans. In practice, clutter, multipath, and hardware differences can introduce **spurious motion responses** that appear as non-human spectral components (Richards et al., 2005). This discrepancy is critical for HPE: when such **spurious motion responses** enter pose reasoning, they can manifest as inter-frame jitter when predictions are evaluated over sequences, thereby limiting temporal stability. This motivates using Doppler signatures in a controlled manner rather than treating them as a symmetric feature stream.

### 2.2. Observation

### 2.2.1. MMWAVE-BASED HUMAN POSE ESTIMATION

**(1) Ignoring Doppler.** Many mmWave HPE models rely predominantly on spatial magnitude, including intensity maps or point clouds (Fan et al., 2024; Zhu et al., 2024).

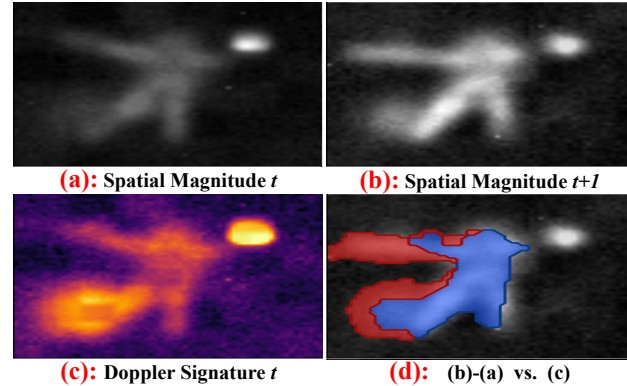

**(a):** Spatial Magnitude *t*  **(b):** Spatial Magnitude *t+1*

**(c):** Doppler Signature *t*  **(d):** (b)-(a) vs. (c)

*Figure 3.* mmWave Dual-Domain Nature. Spatial magnitude in (a) and (b) is obtained by averaging $|\mathbf{H}_t|$ along Doppler. Panel (c) visualizes the Doppler response at $t$. Panel (d) compares inter-frame spatial change with the Doppler pattern. The partial overlap (blue) and distortion (red) between Doppler and spatial variations validate the necessity of selective Doppler prompting. No super-resolution or learned enhancement is applied in this visualization.

While effective for localization, these models do not leverage motion-sensitive Doppler cues and can exhibit inter-frame fluctuations over sequences (Fan et al., 2024).

**(2) Naïve fusion of Doppler and magnitude.** Other methods incorporate Doppler signatures via concatenation or largely symmetric processing with spatial magnitude (Lee et al., 2023; Choi et al., 2025). Such designs do not explicitly account for the heterogeneous reliability of Doppler responses in clutter and multipath environments (Chen et al., 2006) and can introduce nuisance motion responses into localization features, thereby limiting temporal stability.

**(3) Multi-frame aggregation.** A third line of work improves stability by consuming multiple consecutive frames and aggregating information over time (Fan et al., 2024; Kini et al., 2025). These methods can benefit from additional temporal context, but they do not directly address how Doppler should be used when its local responses are unreliable, and they introduce stronger test-time assumptions about access to continuous sequences.

Other designs provide global/local aggregation (Cao et al., 2022), efficient point-cloud pipelines (An & Ogras, 2022), skeleton priors (Tian et al., 2025), generative radar modeling (Huang & McCann, 2025), and egocentric sensing (Li et al., 2023), but they typically introduce extra assumptions, including additional priors or hardware.

Recent complementary directions preserve richer radar structure or exploit physics-driven priors, and operate on 4D radar tensors and therefore assume a richer angular observation than the RAD inputs available in our three benchmarks (Ho et al., 2024), while PPPR (Zheng et al., 2025) and M-GS (Zheng et al., 2026) instead emphasize physics-driven data generation or augmentation. These directions are com-

plementary to PULSE because our contribution focuses on how Doppler cues should guide spatial reasoning once a radar tensor is given.

## 2.3. Summary and Motivation

Our observations can be summarized as: (1) Doppler signatures provide motion-sensitive cues that can support stable prediction but can be unreliable under nuisance factors; and (2) existing mmWave-based approaches either ignore Doppler, fuse it symmetrically with spatial magnitude, or rely on multi-frame aggregation without reliability-aware Doppler screening. These motivate our central hypothesis: **temporally stable mmWave HPE benefits from role-asymmetric, reliability-gated use of Doppler signatures**. Specifically, Doppler should act as a screened motion prompt that conditions spatial magnitude reasoning only when local motion cues are reliable, rather than as a symmetric feature channel.

## 3. Methodology

In this section, we detail Prompting Using Local Spectral Estimates (**PULSE**), which leverages Doppler signatures as controlled cues to address two coupled limitations: sensitivity to spurious motion responses and the temporal instability.

## 3.1. Overview and Problem Formulation

At each time step $t$, a mmWave frame is represented as a three-dimensional tensor $\mathbf{H}_t \in \mathbb{R}^{R \times A \times D}$, where $R$, $A$, and $D$ denote the number of range bins, angle bins, and Doppler bins, respectively. Given a sequence of mmWave frames $\{\mathbf{H}_t\}_{t=1}^{T}$, our objective is to estimate the 3D human poses

$$\mathbf{P}_t \in \mathbb{R}^{J \times 3}, \tag{1}$$

where $J$ is the number of joints. PULSE supports a controllable number of input frames. Fig. 4 shows how Doppler cues in one frame are screened and used in PULSE, which is further extended with a multi-frame mode that aggregates Doppler cues over a short window without changing the spatial backbone (Sec. 3.5).

## 3.2. Dual-Domain Feature Representation

mmWave inherently entangles static spatial structure and dynamic motion effects (Richards et al., 2005). To preserve the distinct physical meanings of these cues, we explicitly decompose $\mathbf{H}_t$ into spatial and Doppler features that share the same lattice while emphasizing complementary aspects.

**Spatial magnitude representation.** The spatial magnitude summarizes range-angle reflectivity, emphasizing spatially coherent scatterers while attenuating transient spectral variations. We compute a magnitude map by averaging responses along the Doppler dimension (Richards et al., 2005):

$$\mathbf{S}_t[r, a] = \frac{1}{D} \sum_{d=1}^{D} |\mathbf{H}_t[r, a, d]|, \quad \mathbf{S}_t \in \mathbb{R}^{R \times A}. \tag{2}$$

This operation marginalizes Doppler variability along the spatial pathway, yielding a representation for localization while preserving Doppler information for prompting.

**Doppler signature representation.** In contrast, Doppler signatures retain a spectral response sensitive to motion, but these cues are not uniquely attributable to articulated body motion (Sec. 2.1.2). We retain the Doppler to form

$$\mathbf{V}_t = |\mathbf{H}_t| \in \mathbb{R}^{R \times A \times D}. \tag{3}$$

For each spatial cell $(r, a)$, the Doppler spectrum

$$\mathbf{v}_{t,r,a} = \mathbf{V}_t[r, a, :] \in \mathbb{R}^{D}, \tag{4}$$

encodes instantaneous motion patterns along the line of sight (Si et al., 2024). These spectra are obtained within the same frame, making them suitable as screening priors.

## 3.3. Tokenization

To enable interaction between the two domains, we convert both into tokens. Importantly, we preserve correspondence between these two tokens and define a locality-constrained interaction neighborhood on the shared lattice, limiting the exposure of each spatial token to spectrally active cells outside its spatial support, thereby avoiding the injection of interference-dominated spectral responses.

**Spatial tokens.** We partition $\mathbf{S}_t$ into non-overlapping patches of size $P_r \times P_a$. Let $\mathbf{S}_t^{(i)} \in \mathbb{R}^{P_r \times P_a}$ denote the $i$-th patch, and $f_s(\cdot)$ be a convolutional patch encoder. Each patch is projected into a $d$-dimensional embedding:

$$\mathbf{t}_i^s = f_s(\mathbf{S}_t^{(i)}) \in \mathbb{R}^d, \qquad \mathbf{T}_s = \{\mathbf{t}_i^s\}_{i=1}^{N_s}, \tag{5}$$

where $N_s = (R/P_r)(A/P_a)$. This patch-level representation balances locality and efficiency, allowing the model to reason about body part structure rather than radar cells.

**Doppler tokens.** For the Doppler cues, each $\mathbf{v}_{t,r,a}$ is independently embedded via an MLP. Let $j \in \{1, \ldots, N_v\}$ index spatial cells on the $R \times A$ grid ($N_v = R \cdot A$), and $\mathbf{v}_{t,j} \in \mathbb{R}^D$ denote the Doppler spectrum at the $j$-th cell. An MLP $f_v : \mathbb{R}^D \to \mathbb{R}^d$ is used to obtain a motion token:

$$\mathbf{t}_{t,j}^v = f_v(\mathbf{v}_{t,j}) \in \mathbb{R}^d. \tag{6}$$

Collecting all cell tokens yields

$$\mathbf{T}_v = \{\mathbf{t}_{t,j}^v\}_{j=1}^{N_v}. \tag{7}$$

This preserves motion at the $R \cdot A$ cell, avoiding early spatial pooling over Doppler. Since $\mathbf{T}_s$ and $\mathbf{T}_v$ are both derived from the same range–angle grid, we can align each spatial token with a local set of Doppler tokens in the prompting.

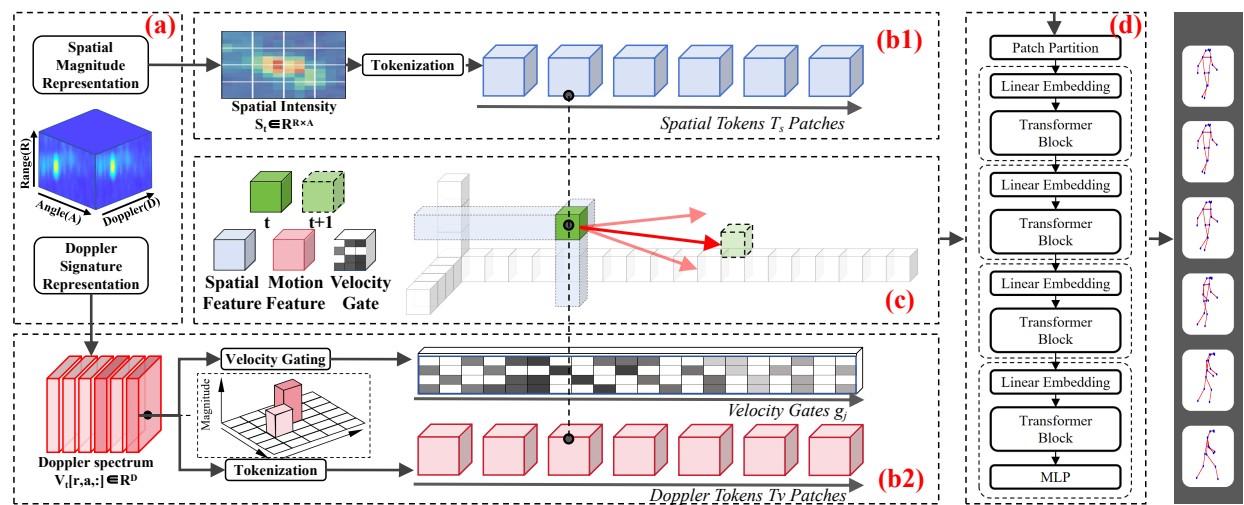

*Figure 4.* Overview of **PULSE**'s core modules (single-frame setting). The pipeline comprises: (a) dual-domain feature construction; (b1) Spatial tokenization and (b2) Doppler tokenization; (c) controlled confidence-aware Doppler prompting; and (d) pose regression.

## 3.4. Controlled Prompting via Conditional Attention

A symmetric combination of spatial magnitude and Doppler signatures can couple localization with nuisance-driven spectral variability (Sec. 2.1.2). PULSE adopts a one-way conditioning mechanism, where Doppler signatures contribute as screened motion cues. The term "prompting" is used to denote this conditional-attention feature modulation.

**Prompt confidence gating.** We first estimate a motion relevance score for each Doppler token:

$$g_{t,j} = \sigma\big(f_g(\mathbf{t}_{t,j}^v)\big), \quad g_{t,j} \in [0,1], \qquad (8)$$

where $f_g$ is a learnable projection and $\sigma(\cdot)$ denotes the sigmoid function. The gate $g_{t,j}$ is learned end-to-end from supervision and modulates how strongly each Doppler token influences spatial reasoning. We treat $g_{t,j}$ as a motion-relevance prior rather than a calibrated signal-quality estimator; additional diagnostics are provided in the Appendix F.2.

**Conditional cross-attention.** Let $\mathbf{t}_i^s$ and $\mathbf{t}_{t,j}^v$ denote spatial and Doppler tokens at time $t$, respectively. We compute motion-conditioned attention weights as

$$\alpha_{ij} = \frac{\exp\left(\frac{(\mathbf{t}_i^s W_Q)(\mathbf{t}_{t,j}^v W_K)^\top}{\sqrt{d_k}} + \beta g_{t,j}\right)}{\sum_{j' \in \mathcal{N}(i)} \exp\left(\frac{(\mathbf{t}_i^s W_Q)(\mathbf{t}_{t,j'}^v W_K)^\top}{\sqrt{d_k}} + \beta g_{t,j'}\right)}, \quad (9)$$

where $W_Q$, $W_K$ are learnable projections, $d_k$ is the key dimension, and $\beta$ controls the influence of motion gating. Note that the query term is fixed to the same spatial token $\mathbf{t}_i^s$ in both the numerator and the denominator; the normalization is taken over Doppler tokens $j' \in \mathcal{N}(i)$.

To reduce cross-region spectral leakage, we do not attend to all Doppler tokens in the full $R \times A$ grid. Instead, for each spatial token $\mathbf{t}_i^s$, we restrict keys/values to a local neighborhood $\mathcal{N}(i)$ on the shared range–angle lattice: $\mathcal{N}(i)$ consists of Doppler tokens whose $(r, a)$ locations fall inside the patch corresponding to $\mathbf{t}_i^s$ and its nearby patches in a fixed window. With patch $P_r \times P_a$, this corresponds to a local receptive field of roughly $(3P_r) \times (3P_a)$ cells. At boundaries, the neighborhood is clipped to a valid index, so each patch attends to all available tokens within the same window. The resulting motion-conditioned context vector is

$$\mathbf{c}_i = \sum_{j \in \mathcal{N}(i)} \alpha_{ij} \left(\mathbf{t}_{t,j}^v W_V\right), \qquad (10)$$

with $W_V$ denoting the value projection. By conditioning attention on $g_{t,j}$, motion cues act as priors that regularize spatial representations rather than competing with them.

## 3.5. Multi-Frame Doppler Prompt Extension

When multiple consecutive frames are available, we extend PULSE by using them only to improve the reliability of Doppler prompting rather than to redesign the spatial backbone. Given a window $\{\mathbf{H}_{t-K+1}, \ldots, \mathbf{H}_t\}$ of $K$ frames, we compute Doppler tokens $\mathbf{t}_j^{v,(\tau)}$ and confidence gates $g_j^{(\tau)}$ for each frame index $\tau$ using the same $f_v$ and $f_g$ as in the single-frame case. We then integrate Doppler tokens at each spatial cell using confidence-weighted aggregation:

$$\bar{\mathbf{t}}_{t,j}^v = \frac{\sum_{\tau=t-K+1}^t g_j^{(\tau)} \mathbf{t}_j^{v,(\tau)}}{\sum_{\tau=t-K+1}^t g_j^{(\tau)} + \epsilon}, \qquad (11)$$

where $\epsilon$ is a small constant. The aggregated Doppler tokens $\bar{\mathbf{T}}_v = \{\bar{\mathbf{t}}_{t,j}^v\}$ preserve the same $R \times A$ lattice and are used in place of $\mathbf{T}_v$ in the prompting stage above. When $K = 1$, this reduces to the single-frame formulation.

### 3.6. Pose Regression

**Motion-conditioned spatial update.** With confidence- and locality-screened conditioning, Doppler signatures act as gated motion cues that refine spatial tokens while limiting nuisance-driven spectral variability. Specifically, each spatial token is updated by a motion-conditioned residual:

$$\tilde{\mathbf{t}}_i^s = \mathbf{t}_i^s + \lambda_i \mathbf{c}_i, \qquad \lambda_i = \sigma(f_\lambda(\mathbf{t}_i^s)), \qquad (12)$$

where $\mathbf{c}_i$ is the motion-conditioned context obtained via gated cross-attention, $f_\lambda(\cdot)$ is a learnable projection, and $\lambda_i \in [0,1]$ controls the extent to which motion cues influence each spatial token. We treat $\beta$ as a global gate-strength hyperparameter, while $\lambda_i$ is a data-dependent coefficient predicted per token, providing token-wise control over motion conditioning. Here, motion cues serve as adaptive, token-wise regularizers that selectively modulate spatial representations based on their motion relevance. This design preserves the primacy of spatial magnitude for pose localization while allowing motion cues to constrain ambiguous or temporally unstable configurations.

**Spatial Reasoning and Regression** The updated spatial tokens $\{\tilde{\mathbf{t}}_i^s\}_{i=1}^{N_s}$ are then processed by stacked transformer layers to model long-range spatial dependencies across body:

$$\mathbf{Z}_t = \text{Transformer}\left(\{\tilde{\mathbf{t}}_i^s\}_{i=1}^{N_s}\right). \qquad (13)$$

Since motion features have already been incorporated as conditioning signals, this module focuses on learning a coherent configuration rather than on reconciling domains.

Finally, joint coordinates are predicted from the fused spatial embedding using a lightweight regression head:

$$\mathbf{P}_t = f_{\text{reg}}(\mathbf{Z}_t), \qquad (14)$$

where $f_{\text{reg}}(\cdot)$ denotes a multilayer perceptron.

## 4. Experiments

### 4.1. Experimental Setup

#### 4.1.1. DATASETS AND PROTOCOLS

We use three public mmWave HPE datasets that differ in radar hardware, human number, and environments: HuPR (Lee et al., 2023), XRF55 (Wang et al., 2024), and mmRadPose (Engel et al., 2025). We focus on datasets that provide motion-sensitive Doppler cues in each frame, enabling consistent testing across methods. Dataset-specific construction and protocol are provided in Appendix C.

HuPR provides approximately 14.1K frames from 6 subjects in a controlled indoor scene and serves as our primary benchmark for studying Doppler behavior under full RAD observations. XRF55 provides approximately 42.9K

frames from 39 subjects across 4 indoor scenes and includes multi-person interactions, making overlapping reflections and clutter more pronounced. mmRadPose provides 432 sequences and approximately 203K frames from 12 subjects performing 12 motion types under three body orientations. It combines an IWR6843AOPEVM radar with OptiTrack motion capture and documents metallic clutter that distorts Doppler-sensitive analysis. Together, these datasets span different hardware, motion diversity, scene layouts, and interference conditions while remaining public benchmarks with aligned 3D supervision.

#### 4.1.2. EVALUATION METRICS

We report four complementary metrics to jointly evaluate per-frame pose accuracy and temporal dynamics fidelity.

**Position accuracy (*mm*):** (1) **MPJPE** (Mean Per Joint Position Error, ↓) as the key measure of 3D joint localization accuracy. (2) **PA-MPJPE** (Procrustes-Aligned MPJPE, ↓), which evaluates pose quality after rigid alignment.

**Temporal dynamics (*mm/frame*):** To characterize temporal behavior, we therefore report two temporal metrics, which are commonly used in vision methods but largely unexplored in mmWave HPE. (1) **MPJVE** (Mean Per Joint Velocity Error, ↓) measures discrepancies between predicted and ground-truth joint velocities, directly reflecting whether the estimated motion follows the true temporal dynamics. (2) **AKV** (Average Keypoint Velocity, ↓) quantifies the average magnitude of predicted joint velocities and has been used in prior work as an indicator of motion smoothness (Fan et al., 2024). While AKV captures the overall intensity of predicted motion, it does not assess whether such motion is physically correct, as overly smooth or nearly static predictions may yield low AKV despite deviating from true dynamics. MPJVE complements this limitation by explicitly comparing predicted and ground-truth velocities, and serves as a practical proxy for *sensitivity to spurious motion responses*: when nuisance-dominated Doppler signatures leak into pose reasoning, they can manifest as jittery velocity estimates even if the per-frame positions remain plausible.

All metrics are computed in the same 3D coordinate system as the dataset annotations. When a dataset provides poses in meters, we convert them to millimeters before computing all metrics, so that all reported numbers use consistent units.

#### 4.1.3. BASELINES

We group the mmWave baselines by original input assumptions. All baselines are evaluated without tracking or post-hoc smoothing. For fairness, **all** multi-frame comparisons use the same window length $K$ during training and inference; unless otherwise stated, we use $K=9$ for all multi-frame methods (mmDiff, milliMamba, and PULSE (KF)).

**Single-frame baselines.** **(1) HuprModel** (Lee et al., 2023): A multi-scale CNN with hierarchical feature pyramids. **(2) MvDoppler** (Choi et al., 2025): A dual-signal method that combines magnitude and Doppler streams. **(3) RETR** (Yataka et al., 2024): A retrieval-based mmWave baseline. **Multi-frame baselines.** These methods explicitly consume multiple consecutive frames at test time. **(4) mmDiff** (Fan et al., 2024): A diffusion model that uses neighboring frames. **(5) milliMamba** (Kini et al., 2025): A multi-frame Mamba-based fusion method.

### 4.1.4. IMPLEMENTATION DETAILS

*Table 1.* Single-Frame (SF) and Multi-Frame (MF) evaluation.

| Dataset | Input | Method | MPJPE↓ | PA-MPJPE↓ | MPJVE↓ | AKV↓ |
|---|---|---|---|---|---|---|
| **HuPR** | SF | HuPRModel | 65.37 | 58.11 | 14.70 | 14.1 |
| | | MvDoppler | 69.71 | 65.56 | 13.11 | 13.4 |
| | | RETR | 78.09 | 72.54 | 15.07 | 16.10 |
| | | **PULSE (1F)** | **60.57** | **54.15** | **9.78** | **5.1** |
| | MF | mmDiff | 65.54 | 60.02 | 13.60 | 5.7 |
| | | milliMamba | 64.08 | 57.44 | 11.69 | 7.8 |
| | | **PULSE (KF)** | **58.64** | **53.01** | **8.16** | **5.0** |
| **XRF55** | SF | HuPRModel | 80.26 | 76.06 | 19.58 | 20.7 |
| | | MvDoppler | 75.11 | 70.25 | 18.16 | 15.4 |
| | | RETR | 81.77 | 78.90 | 19.61 | 16.54 |
| | | **PULSE (1F)** | **70.34** | **67.51** | **15.33** | **7.1** |
| | MF | mmDiff | 79.06 | 74.31 | 21.44 | 9.2 |
| | | milliMamba | 74.35 | 69.33 | 17.76 | 12.37 |
| | | **PULSE (KF)** | **68.99** | **63.81** | **14.05** | **6.5** |
| **mmRad Pose** | SF | HuPRModel | 77.84 | 70.33 | 15.02 | 17.7 |
| | | MvDoppler | 74.17 | 67.50 | 14.28 | 13.2 |
| | | RETR | 78.09 | 72.31 | 15.34 | 16.87 |
| | | **PULSE (1F)** | **68.83** | **60.80** | **12.90** | **6.5** |
| | MF | mmDiff | 76.31 | 69.04 | 17.31 | 7.6 |
| | | milliMamba | 72.34 | 65.77 | 14.26 | 12.4 |
| | | **PULSE (KF)** | **67.56** | **59.19** | **11.7** | **5.4** |

### 4.2. Single-Person Evaluation

Table 1 shows the quantitative results. Single-Frame mode PULSE (1F) consistently reduces MPJVE across all datasets, indicating improved motion-dynamics fidelity. Notably, on HuPR, PULSE (1F) achieves the lowest MPJVE among the reported methods, including the multi-frame baseline mmDiff (Fan et al., 2024); the frame-wise comparison in Fig. 5 further illustrates that mmDiff exhibits larger velocity deviations from the ground truth on representative sequences. The reference curve in the bottom panel of Fig. 5 represents the ground-truth joint velocity magnitude rather than a velocity error and is used only to show the scale of the underlying motion intensity. Across datasets, reductions in AKV are accompanied by reductions in MPJVE, suggesting that stability gains are not explained by suppressing motion but by aligning dynamics more closely with natural trajectories.

Beyond PULSE (1F), some downstream applications can provide input sequences, so we further evaluate PULSE

in Multi-Frame (MF) scenarios. PULSE's MF mode integrates Doppler-branch outputs from consecutive frames before prompting, which further improves stability beyond PULSE (1F). Importantly, these cross-dataset gains provide empirical evidence consistent with PULSE attenuating nuisance-driven Doppler cues induced by environmental reflections and hardware artifacts.

*Table 2.* Multi-person benchmark on XRF55.

| Dataset | Input | Method | MPJPE↓ | PA-MPJPE↓ | MPJVE↓ | AKV↓ |
|---|---|---|---|---|---|---|
| **XRF55** | SF | HuPRModel | 86.20 | 82.05 | 25.19 | 37.9 |
| | | MvDoppler | 83.64 | 79.33 | 22.47 | 18.7 |
| | | RETR | 86.14 | 83.06 | 27.50 | 36.7 |
| | | **PULSE (1F)** | **73.59** | **70.04** | **18.80** | **9.2** |
| | MF | mmDiff | 83.57 | 80.01 | 27.08 | 13.7 |
| | | milliMamba | 79.71 | 73.15 | 20.53 | 11.9 |
| | | **PULSE (KF)** | **72.17** | **68.59** | **16.34** | **8.0** |

### 4.3. Multi-Person Evaluation

We further evaluate performance in multi-person scenarios on the XRF55. In multi-person scenes, the spatial grid contains overlapping reflections from multiple bodies (Sec. 2.1.2), making this a stringent test of whether Doppler cues can be incorporated without amplifying interference-driven artifacts. PULSE is designed to mitigate this via locality-restricted prompting over $\mathcal{N}(i)$ and confidence gating with $g_{t,j}$, which together aim to downweight spectrally active but pose-irrelevant Doppler cues.

Table 2 reports the results. Two trends can be observed: (1) all baselines exhibit clear performance drops in multi-person settings, confirming that multi-person mmWave HPE remains a challenge. (2) Despite this increased difficulty, our method exhibits notably smaller degradation in all metrics.

PULSE (1F) achieves lower errors than single-frame baselines, indicating improved temporal stability under inter-person interference. The multi-frame mode (PULSE (KF)) further improves stability when short windows are available by increasing prompt reliability through confidence-weighted aggregation. We follow the dataset protocol for multi-person scenes, predictions are evaluated using the standard association procedure (Appendix C), without introducing additional tracking, or post-hoc smoothing.

These results are consistent with our design goal of reducing the impact of nuisance Doppler cues. In particular, the ablations that remove motion gating or patch-level locality (Table 5) support their role in stabilizing predictions.

### 4.4. PULSE Prompting as a Plug-in

To probe whether PULSE's core contribution is beneficial beyond our own regressor, we plug PULSE into two multi-frame backbones, mmDiff (Fan et al., 2024) and milli-

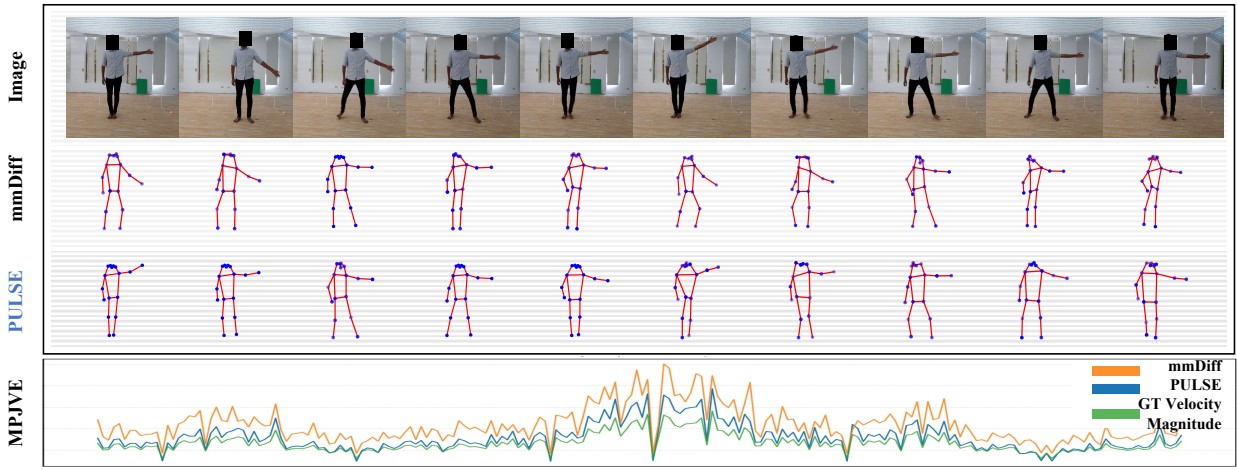

*Figure 5.* Qualitative comparison on HuPR. **Top:** Pose predictions of mmDiff and PULSE on a continuous sequence. **Bottom:** Frame-wise MPJVE curves for mmDiff and our method, together with the ground-truth velocity magnitude as a reference scale for motion intensity.

Mamba (Kini et al., 2025). Concretely, we keep the baseline backbone and prediction head unchanged and replace only the front-end fusion stage with PULSE prompting, and feed the prompted features into the original backbone. All plug-in experiments use the same multi-frame window length in Sec 4.3. Table 3 summarizes results on three datasets. Plugging PULSE into two backbones improves both MPJPE and MPJVE across datasets, indicating that controlled Doppler prompting can serve as a drop-in replacement for naïve fusion in existing multi-frame architectures.

*Table 3.* PULSE as a plug-in on mmDiff(A) and millimamba(B).

| Backbone | HuPR | | XRF55 | | mmRadPose | |
| | MPJPE↓ | MPJVE↓ | MPJPE↓ | MPJVE↓ | MPJPE↓ | MPJVE↓ |
|---|---|---|---|---|---|---|
| (A) orig. | 65.54 | 13.60 | 79.06 | 21.44 | 76.31 | 17.31 |
| (A) + PULSE | **60.89** | **10.07** | **72.56** | **16.02** | **70.59** | **13.27** |
| (B) orig. | 68.08 | 11.69 | 74.35 | 17.76 | 72.34 | 14.26 |
| (B) + PULSE | **63.43** | **10.47** | **68.25** | **14.37** | **66.54** | **10.49** |

### 4.5. Ablation Studies

Unless otherwise stated, ablations are performed on HuPR and evaluated in the **single-frame (1F)** setting, avoiding conflating the effects of multi-frame temporal aggregation with the behavior of the controlled Doppler prompting modules that constitute our core contribution. Additional sensitivity analyses on gate supervision, neighborhood size, and spatial patch size are reported in Appendix F. These analyses complement the core ablations in this section and provide practical guidance for adapting PULSE to different radar resolutions.

#### 4.5.1. ABLATING DUAL-DOMAIN MODELING

This ablation examines whether explicitly separating spatial magnitude and Doppler signatures is essential. To ablate

*Table 4.* Ablation on dual-domain modeling.

| Variant | MPJPE↓ | PA-MPJPE↓ | MPJVE↓ | AKV↓ |
|---|---|---|---|---|
| Spatial-only | 69.05 | 64.81 | 12.54 | 14.8 |
| Doppler-only | 76.44 | 72.17 | 28.07 | 21.8 |
| **Full** | **60.57** | **54.15** | **9.78** | **5.1** |

a branch, we disable its contribution by setting its feature output to zeros (equivalently, removing that pathway) while keeping the rest of the architecture unchanged. Table 4 confirms a clear role separation between the two domains. The spatial-only variant attains moderate pose accuracy but exhibits **severe instability**, indicating large velocity magnitudes and pronounced jitter when the Doppler-motion pathway is disabled. In contrast, the Doppler-only variant fails to localize structure and also becomes dynamically unreliable, suggesting that motion signatures alone cannot resolve pose geometry. The full model achieves the best of both position and velocity accuracy, showing that explicit dual-domain modeling is necessary to jointly achieve accurate localization and physically plausible, stable motion.

#### 4.5.2. ABLATING PATCH-WISE ALIGNMENT AND MOTION-GUIDED INTERACTION

To verify that the observed performance gains stem from our proposed design rather than increased model capacity, we evaluate three control variants. In all cases, the regression backbone remains identical: **(1) Direct Concatenation**: Spatial and locally pooled Doppler tokens are concatenated and processed jointly, bypassing any conditional attention or gating mechanisms. **(2) Ungated Cross-Attention**: The conditional cross-attention module is retained, but the motion gate is ablated from the attention logits. Consequently, Doppler tokens are aggregated exclusively based on con-

*Table 5.* Ablation on alignment and motion-guided interaction.

| Variant | MPJPE↓ | PA-MPJPE↓ | MPJVE↓ | AKV↓ |
|---|---|---|---|---|
| w/ naïve concat | 68.14 | 67.51 | 15.33 | 17.9 |
| w/ ungated cross-attn | 67.68 | 65.09 | 13.89 | 14.1 |
| w/o patch-level align | 67.13 | 63.46 | 13.07 | 10.4 |
| w/o motion gating | 65.09 | 60.49 | 12.17 | 11.9 |
| **Full** | **60.57** | **54.15** | **9.78** | **5.1** |

tent similarity. **(3) Global Interaction**: Local patch-level alignment is replaced by a global receptive field, exposing each spatial token to cross-region Doppler responses without neighborhood constraints.

As shown in Table 5, removing patch-level alignment significantly degrades stability, consistent with the injection of motion cues at incorrect spatial locations. Removing motion gating $g_{tj}$ similarly increases sensitivity to clutter-induced Doppler noise, reflecting over-reactive dynamics. In contrast, the full model restores both accuracy and stability, empirically justifying our design.

## 5. Conclusion

We study mmWave-based HPE and propose **PULSE**, which treats Doppler signatures as controlled prompts to condition spatial magnitude reasoning. Experiments across three public datasets, including single- and multi-person settings, demonstrate consistent improvements in per-frame accuracy and better temporal metrics under frame-wise evaluation. In the future, to more directly stress-test and refine PULSE's key components (e.g., gating, locality, and prompt aggregation), we plan to collect targeted benchmark data spanning diverse activity primitives and controlled environmental perturbations, enabling systematic evaluation and optimization under controlled sources of spurious Doppler responses.

## Acknowledgements

This work was supported by the Department of Computer Science at the University of Warwick. It is also partially supported by the National Natural Science Foundation of China (Grant No. 62502040), and the Postdoctoral Fellowship Program of the China Postdoctoral Science Foundation (Grant No. GZC20251056).

## Impact Statement

This work advances single-frame human pose estimation using millimeter-wave radar, with a particular emphasis on temporal stability—an important requirement for safety- and health-oriented monitoring where unstable trajectories can lead to unreliable downstream decisions. A primary motivation of this study is to support privacy-preserving and non-intrusive sensing in settings such as clinical monitoring, assisted living, and long-term behavioral analysis, where camera-based systems may be impractical or undesirable. By operating without capturing identifiable visual data, mmWave-based approaches can reduce privacy risks compared to vision-based alternatives. At the same time, like other sensing and monitoring technologies, the proposed approach could be misused for unauthorized surveillance if deployed without appropriate safeguards. Such risks are not unique to this work and are largely determined by deployment context rather than algorithmic design. We believe that responsible use, informed consent, and adherence to applicable regulations are essential for mitigating potential negative impacts. By improving temporal stability under single-frame sensing, the proposed approach may reduce nuisance-driven fluctuations that could otherwise contribute to false alarms or inconsistent behavioral measurements in long-term monitoring.

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

## A. Experimental Configuration and Efficiency Analysis

### A.1. Hardware and Software Environment

All experiments are conducted on a single workstation with the following configuration: **CPU:** Intel Core i9-13900K,

**GPU:** NVIDIA RTX 4060 (8GB VRAM), **Memory:** 32 GB DDR5 RAM, **Operating System:** Windows 11, **CUDA:** CUDA 12.1. All models are implemented in PyTorch and evaluated using official or author-released codebases whenever available.

## A.2. Key Hyperparameters for Reproducibility

Table 6 summarizes the main hyperparameters.

| Category | Setting |
|---|---|
| Range–angle resolution | $R \times A = 64 \times 64$ (after resampling) |
| Doppler bins | $D = 16$ |
| Spatial patch size | $P_r \times P_a = 4 \times 4$ |
| Spatial tokens | $N_s = (R/P_r)(A/P_a) = 256$ |
| Doppler tokens | $N_v = R \cdot A = 4096$ (cell-level) |
| Embedding dimension | $d = 32$ |
| Transformer depth | 4 layers |
| Attention heads | 4 heads |
| Dropout | 0.1 |
| Cross-attention locality | $3 \times 3$ patch neighborhood $\mathcal{N}(i)$ |
| Gate strength | $\beta = 1$ |
| Multi-frame window length | $K = 9$ (Protocol in milliMamba) |
| Optimizer | Adam |
| Learning rate | $1 \times 10^{-4}$ |
| Weight decay | 0.01 |
| Batch size | 8 |
| Epochs | 100 |
| Gradient clipping | 1.0 (global norm) |

*Table 6.* Key model and training hyperparameters.

## A.3. Efficiency Measurement Protocol

To ensure fair comparison, we adopt a unified evaluation protocol: **Input size** refers to the per-frame input tensor resolution used at inference time. **Latency** is measured as the average inference time per frame, excluding data loading and preprocessing, averaged over 1,000 forward passes with batch size 1. **#Params** denotes the total number of trainable parameters. **MFLOPs** are computed for a single forward pass under the corresponding input resolution. All latency measurements are performed with GPU synchronization enabled to avoid underestimation.

## A.4. Runtime Efficiency Comparison

Table 7 summarizes the computational characteristics of all evaluated methods.

Several observations can be drawn from Table 7. The existing mmWave-based methods exhibit lower input dimensionality, but many still adopt deep architectures that treat Doppler information as generic feature channels.

In contrast, our method achieves a favorable balance between accuracy, temporal stability, and computational efficiency. By leveraging motion-relevant Doppler cues at the

| Method | Input Size | Latency (ms) | #Params (M) | FLOPs (M) |
|---|---|---|---|---|
| HuprModel | 64*64*16 | 27.1 | 324.9 | 254 |
| MvDoppler | 64*64*16 | 7.6 | 36.7 | 164 |
| RETR | 64*64*16 | 11.5 | 76.9 | 164 |
| **PULSE(1F)** | 64*64*16 | 5.1 | 12.0 | 75 |
| mmDiff | $K \times$point | 19.9 | 182.8 | 264 |
| millimamba | $K \times$point | 14.1 | 32.8 | 121 |
| **PULSE(KF)** | $K \times$point | 7.2 | 12.3 | 93.4 |

*Table 7.* Computational efficiency comparison of mmWave-based methods on HuPR dataset.

representation level, we improve stability metrics without post-hoc temporal smoothing; when short windows are available, our multi-frame extension can further enhance prompt reliability. As a result, our model remains lightweight while yielding more coherent trajectories under frame-wise evaluation.

## B. Adaptation of mmWave-Based Baselines

To enable fair and informative comparison, we include several representative mmWave-based human pose estimation methods as baselines. All mmWave baselines are evaluated under clearly specified input assumptions, and their adaptations are kept minimal, preserving the original modeling intent while aligning with our evaluation protocol. We avoid any post-hoc temporal smoothing so that temporal stability differences primarily reflect how each method handles motion-sensitive cues under its test-time input assumption. For HuprModel (Lee et al., 2023), and MvDoppler (Choi et al., 2025), which originally use multi-view inputs, we remove multi-view aggregation modules and retain their core single-view processing and fusion mechanisms; for milliMamba (Kini et al., 2025), we follow the paper description due to the absence of an official release.

**mmDiff** (Fan et al., 2024) is selected because it is one of the few existing works that explicitly target temporal stability in mmWave-based pose estimation. The method employs a diffusion-based denoising process on aggregated radar point clouds and relies on concatenating multiple adjacent frames to mitigate point sparsity. We report mmDiff as a multi-frame baseline and evaluate it using the same metrics (MPJPE/PA-MPJPE/MPJVE/AKV) without post-hoc filtering.

**HuprModel** (Lee et al., 2023) is chosen due to its modeling of temporal features in mmWave data. The original HuPRModelframework exploits multi-view radar inputs and hierarchical feature pyramids. For fair comparison, we remove the multi-view fusion components and retain the core spatial–temporal processing modules operating on single-view inputs. This adaptation preserves HuprModel's temporal feature extraction mechanism while avoiding the need for additional information from multiple viewpoints. Com-

paring against HuPRModelallows us to examine whether generic temporal feature encoding alone is sufficient for stable pose estimation, in contrast to our motion-conditioned fusion strategy.

**MvDoppler** (Choi et al., 2025) is included because it explicitly distinguishes between positional and velocity-related information, treating them as two separate signal modalities. However, in MvDoppler, these modalities undergo largely naïve processing pipelines with similar architectural structures. We remove the multi-view aggregation modules and retain the core dual-modal fusion mechanism. This baseline is particularly relevant for evaluating whether simply separating magnitude and Doppler signals is sufficient, or whether Doppler features require a specialized, motion-aware role. Performance differences between MvDoppler and our approach directly reflect the benefit of our asymmetric, motion-guided treatment of Doppler cues.

**RETR** (Yataka et al., 2024) is included as a single-frame baseline that relies on retrieval-based matching rather than explicit Doppler prompting. We follow the authors' reported preprocessing and model configuration and evaluate it under the same single-frame protocol.

**milliMamba** (Kini et al., 2025) is included as a representative recent multi-frame baseline that models spatio-temporal dependencies across consecutive radar frames. Since its official implementation has not been released, we reimplement it based on the architecture and hyperparameters described in the paper. In our experiments, we follow the authors' default setting and use a fixed window of $K=9$ consecutive frames during both training and inference.

We validated our reimplementation by matching the reported architectural configuration and confirming that our reproduced results on the datasets used in this paper (mainly the HuPR dataset (Lee et al., 2023)) fall within the same performance range as those reported in the original work.

Overall, these baselines span diffusion-based temporal aggregation, temporal feature encoding, and dual-modal fusion paradigms. By adapting each method in a controlled and transparent manner, we are able to isolate and validate the specific advantages of our proposed dual-domain, motion-conditioned design across complementary modeling choices.

# C. Dataset-Specific Input Construction and Normalization

Our framework assumes an idealized mmWave representation tion $\mathbf{H}_t \in \mathbb{R}^{R \times A \times D}$, in which spatial dimensions encode geometric structure and the Doppler dimension captures instantaneous motion cues. Importantly, this formulation does not require a physically complete or invertible Doppler spec-

tra. Throughout this work, Doppler information is treated as a motion-aware prior that regularizes spatial representations, rather than as an independent modality that needs to be fully reconstructed.

In practice, publicly available mmWave HPE datasets exhibit substantial heterogeneity in radar hardware, signal-processing pipelines, and Doppler availability. This appendix clarifies how each dataset is mapped into a unified input format, how dataset-specific constraints are respected, and how fair comparisons are maintained across baselines.

We evaluate on three public mmWave HPE datasets: HuPR (Lee et al., 2023), XRF55 (Wang et al., 2024), and mmRadPose (Engel et al., 2025). They differ in (i) environment and subject configuration (single-/multi-person), (ii) radar preprocessing and provided representations, and (iii) pose acquisition pipelines. HuPR and mmRadPose provide per-frame range–angle–Doppler tensors, while XRF55 provides RA and RD maps from which we reconstruct a RAD tensor. Ground-truth 3D poses are provided by the dataset authors via synchronized sensing and calibration (HuPR/XRF55) or a motion-capture system (mmRadPose). Table 8 summarizes the provided mmWave representations and evaluation settings across the three datasets.

| Dataset | Provided mmWave | Setting |
|---------|-----------------|---------|
| XRF55 | RA/RD | multi-person |
| HuPR | R-A-D | single-person |
| mmRadPose | R-A-D | single-person |

*Table 8.* Summary of the evaluated mmWave HPE datasets.

We select these datasets because they expose motion-sensitive Doppler information in one mmWave frame (or can be mapped to our unified RAD lattice) and provide 3D pose supervision, enabling controlled study of how nuisance Doppler responses affect stability under frame-wise evaluation. Together, they span heterogeneous sensing and preprocessing conditions and include both single- and multi-person settings, which helps stress-test robustness to environment- and interference-driven Doppler artifacts without changing the model interface.

## C.1. Dataset Diversity and Motivation

Unlike prior mmWave HPE studies that evaluate on one or two datasets (Lee et al., 2023; Fan et al., 2024; Zhu et al., 2024; Choi et al., 2025; Xue et al., 2023; Cui & Dahnoun, 2021; Sengupta & Cao, 2022; Sengupta et al., 2020), we include multiple widely used datasets that provide aligned mmWave measurements and 3D pose supervision. This choice is motivated by the sensitivity of mmWave signals to environment layout, sensor configuration, and preprocessing choices, and it enables a broader assessment across data formats and sensing conditions.

The datasets used in this work differ along three key axes: (i) radar configuration and antenna layout, (ii) availability and fidelity of Doppler information, and (iii) preprocessing pipelines that may compress or marginalize Doppler content. Rather than enforcing a uniform reconstruction of Doppler spectra, we respect these dataset-specific constraints and avoid hallucinating missing motion information.

### C.2. Pose Supervision and Fairness

The datasets used in this work provide 3D human pose supervision through established sensing pipelines. HuPR, XRF55 rely on synchronized vision or depth sensors combined with calibration and post-processing to obtain 3D joint annotations aligned with mmWave measurements, as released by the dataset authors (Lee et al., 2023; Wang et al., 2024). mmRadPose (Engel et al., 2025), in contrast, provides 3D poses captured using a dedicated motion capture system, which is widely recognized as the gold standard for precise human motion analysis, yielding high-precision joint trajectories and serving as a complementary benchmark under high-fidelity motion supervision. Across all datasets, we strictly use the released annotations, official splits, and dataset-recommended protocols. No additional temporal smoothing, cross-frame aggregation, or post-processing is applied when constructing supervision, ensuring that all compared methods are trained and evaluated under identical conditions within each dataset.

### C.3. Input Construction per Dataset

**Single-Frame Doppler Clarification**   In FMCW radar, Doppler information is extracted via Fourier analysis across chirps **within the same frame**. All Doppler representations in this work are constructed strictly from measurements inside frame $t$. We do not compute temporal differences or statistics across adjacent frames, and there is no multi-frame leakage in our single-frame setting. When we evaluate multi-frame methods (including PULSE (KF)), the input is a short window $\{\mathbf{H}_{t-K+1}, \ldots, \mathbf{H}_t\}$ with a fixed $K{=}9$ (Kini et al., 2025). In this setting, Doppler spectra are still computed within each frame, and the only cross-frame operation is confidence-weighted aggregation of Doppler prompts prior to prompting; the spatial magnitude pathway remains frame-local.

**XRF55**   For XRF55, the released measurements are two 2D maps (RA and RD); RA controls the angular energy distribution, while RD (normalized per range) provides a Doppler distribution. We reconstruct a per-frame 3D RAD tensor by distributing the per-range Doppler profile over angles using the RA energy weights. Let $\mathbf{M}_t^{RA}[r, a] \in \mathbb{R}$ denote the RA map and $\mathbf{M}_t^{RD}[r, d] \in \mathbb{R}$ denote the RD map.

We first form per-range angular weights

$$w_t[r, a] = \frac{\mathbf{M}_t^{RA}[r, a]}{\sum_{a'=1}^{A} \mathbf{M}_t^{RA}[r, a'] + \epsilon}, \qquad (15)$$

and then construct the RAD tensor as

$$\mathbf{H}_t[r, a, d] = w_t[r, a] \cdot \mathbf{M}_t^{RD}[r, d], \qquad (16)$$

where $\epsilon$ is a small constant. This construction preserves the released RD distribution when marginalizing over angle, while using RA to allocate Doppler energy across angles.

For fairness, all mmWave-based methods on XRF55 are provided with the same constructed RAD tensor or corresponding point cloud. No baseline is given access to extra dataset-specific annotations beyond those provided by XRF55. This ensures that performance differences on XRF55 arise from how models exploit spatial and Doppler cues, rather than from unequal input representations or preprocessing advantages.

**HuPR and mmRadPose**   HuPR and mmRadPose provide explicit range–angle–Doppler information with minimal compression. For these datasets, we directly construct $\mathbf{H}_t \in \mathbb{R}^{R \times A \times D}$ using standard FMCW processing. They therefore serve as primary benchmarks for evaluating the role of Doppler signatures.

### C.4. Multi-Person Handling

For datasets containing multiple subjects (XRF55), our framework predicts a single pose per detected person. Matching between predictions and ground truth is performed using the standard closest-distance association in 3D joint space, consistent with prior mmWave HPE work (Yataka et al., 2024; Choi et al., 2025). This design isolates pose estimation from detection ambiguity and allows fair evaluation under both single- and multi-person settings.

### C.5. Normalization and Alignment

Radar intensities are normalized **per frame** to mitigate hardware-dependent scale variations without introducing sequence-level statistics. Doppler-related features are normalized independently and treated as dataset-internal motion descriptors; we do not attempt to align Doppler bins to a common physical velocity unit across datasets.

All 3D poses are expressed in a pelvis-centered coordinate system with consistent joint ordering and scale normalization, ensuring that MPJPE, MPJVE, and AKV are comparable across datasets.

## D. Training Protocols and Reliability

### D.1. Training Strategy

PULSE is trained in a supervised manner using paired mmWave inputs and 3D joint annotations. The primary objective is the per-frame joint regression loss:

$$\mathcal{L}_{\text{pos}} = \frac{1}{J} \sum_{j=1}^{J} \|\hat{\mathbf{p}}_{t,j} - \mathbf{p}_{t,j}\|_2, \qquad (17)$$

where $\hat{\mathbf{p}}_{t,j}$ and $\mathbf{p}_{t,j}$ denote the predicted and ground-truth coordinates of joint $j$ at time $t$, respectively.

No explicit temporal loss (e.g., velocity or smoothness regularization) is applied during training. In the single-frame setting, temporal stability emerges from Doppler motion cues contained within each frame. In the multi-frame setting, the model uses a short window only to aggregate Doppler prompts, and the loss remains unchanged. This design ensures that improvements in MPJVE and AKV reflect better motion modeling rather than post-hoc smoothing.

Baselines are trained using their official protocols when available; otherwise, we follow the reported settings and match input formats as described in Appendix B. PULSE is trained using the Adam optimizer with a fixed learning rate, and early stopping is applied based on validation MPJPE. Detailed hyperparameters, including learning rates, batch sizes, and training epochs for each dataset, are reported in the supplementary material.

### D.2. Temporal Metrics and Coordinate Conventions

We clarify the computation of velocity-based evaluation metrics used throughout the paper.

**Velocity Definition** Joint velocities are computed using first-order finite differences between consecutive frames:

$$\hat{\mathbf{v}}_{t,j} = \frac{\hat{\mathbf{p}}_{t+1,j} - \hat{\mathbf{p}}_{t,j}}{\Delta t}, \quad \mathbf{v}_{t,j} = \frac{\mathbf{p}_{t+1,j} - \mathbf{p}_{t,j}}{\Delta t}. \quad (18)$$

Unless otherwise specified, we set $\Delta t = 1$, corresponding to per-frame differences at the native dataset frame rate. No temporal smoothing, filtering, or interpolation is applied to predicted or ground-truth poses.

We report MPJVE and AKV in **mm/frame** under $\Delta t = 1$. For consistency across datasets, we convert pose coordinates to millimeters (if originally provided in meters) before computing both position- and velocity-based metrics.

**MPJVE** Mean Per Joint Velocity Error (MPJVE) measures the discrepancy between predicted and ground-truth joint velocities:

$$\text{MPJVE} = \frac{1}{(T-1)J} \sum_{t=1}^{T-1} \sum_{j=1}^{J} \|\hat{\mathbf{v}}_{t,j} - \mathbf{v}_{t,j}\|_2. \quad (19)$$

In mmWave HPE, elevated velocity errors can be symptomatic of nuisance-driven spectral variability being interpreted as motion; thus, MPJVE serves as a practical indicator of sensitivity to spurious motion responses and their downstream impact on temporal stability.

**AKV** Average Keypoint Velocity (AKV) measures the average magnitude of predicted joint velocities:

$$\text{AKV} = \frac{1}{(T-1)J} \sum_{t=1}^{T-1} \sum_{j=1}^{J} \|\hat{\mathbf{v}}_{t,j}\|_2. \quad (20)$$

AKV reflects the overall motion intensity of predictions and is reported alongside MPJVE to distinguish physically plausible motion from over-smoothed or nearly static outputs.

### D.3. Handling Dataset Heterogeneity

Datasets differ in resolution, angular coverage, and availability of Doppler information. To ensure fair comparison, we apply the following normalization principles:

**Unified input resolution.** All inputs are resampled to a fixed range–angle grid before tokenization using bilinear interpolation on the $R \times A$ plane. This step ensures that architectural capacity and receptive fields remain comparable across datasets.

**Consistent pose parameterization.** All predicted poses are represented using the same joint topology and coordinate system. When necessary, dataset-specific joint annotations are mapped to a common skeleton following standard conventions (Engel et al., 2025).

**No dataset-specific tuning.** Apart from unavoidable input adaptation described in Appendix A, no dataset-specific architectural changes or loss reweighting are introduced. The same model configuration is used across all benchmarks.

### D.4. Multi-Person Training and Evaluation Consistency

For datasets containing multiple subjects, we strictly follow the dataset-provided training and evaluation protocols (Wang et al., 2024). Specifically, all baselines use the same instance definitions, ground-truth associations, and evaluation splits released by the dataset authors. No additional person detection, tracking, or cross-frame association is introduced by our method. During training and inference, each annotated person instance is processed independently according to the dataset protocol.

### D.5. Baseline Reproduction and Fairness

For mmWave-based baselines, we follow the official training protocols whenever code or pretrained models are available. When reimplementation is required, we match input resolution, training epochs, and backbone capacity to the extent

possible. For multi-frame methods, we fix the number of input frames to $K=9$ during both training and inference, and report them as multi-frame baselines. All baselines are evaluated using the same data splits and evaluation metrics. Velocity-based metrics (MPJVE and AKV) are computed from predicted trajectories without temporal filtering or post-processing.

## E. Cross-Dataset Generalization

To evaluate robustness beyond in-domain settings, we conduct a cross-dataset generalization experiment. Specifically, we train models on HuPR and directly evaluate them on mmRadPose without any fine-tuning. This setting tests generalization across different radar hardware, antenna configurations, Doppler resolutions, and capture environments, and directly addresses concerns about potential overfitting to dataset-specific Doppler scales or array geometry.

All models are trained on the HuPR training split using identical protocols. During testing on mmRadPose, inputs are adapted only through the same range–angle resampling; no dataset-specific normalization, temporal alignment, or scale calibration is applied.

Table 9 reports MPJPE and MPJVE on mmRadPose. Across all methods, performance degrades compared to in-dataset evaluation, reflecting the inherent difficulty of cross-sensor/cross-scene transfer. However, methods that couple motion cues without reliability screening can degrade more in MPJVE under domain shift. In contrast, PULSE (1F) shows more graceful degradation in both spatial accuracy and temporal consistency.

This supports our design choice of treating Doppler cues as adaptive regularizers rather than as rigid geometric constraints, enabling improved robustness under domain shift.

*Table 9.* Cross-dataset generalization (HuPR→mmRadPose). mmDiff is evaluated with $K=9$ input frames.

| Method | MPJPE ↓ | MPJVE ↓ |
|---|---|---|
| HuPRModel (Lee et al., 2023) | 85.42 | 27.01 |
| MvDoppler (Choi et al., 2025) | 77.09 | 17.63 |
| mmDiff ($K=9$) (Fan et al., 2024) | 83.64 | 23.71 |
| **PULSE (1F)** | 70.05 | 14.36 |

## F. More Ablations

Unless otherwise specified, analyses in this section are conducted under the **single-frame (1F)** setting on HuPR.

### F.1. Comparison with Post-Hoc Kalman Filtering

To separate selective motion conditioning from generic trajectory smoothing, we apply a standard constant-velocity

Kalman filter to the frame-wise predictions of HuPRModel on the HuPR test set. The filter reduces AKV and modestly lowers MPJPE, but it increases MPJVE because genuine rapid motion is attenuated together with jitter. PULSE improves both MPJPE and MPJVE without post-hoc filtering, which is consistent with suppressing nuisance-driven motion cues before pose regression.

*Table 10.* Comparison with post-hoc Kalman filtering on HuPR.

| Method | MPJPE ↓ | MPJVE ↓ | AKV ↓ |
|---|---|---|---|
| HuPRModel | 65.37 | 14.70 | 14.1 |
| HuPRModel + Kalman | 63.50 | 15.60 | 9.8 |
| PULSE (1F) | **60.57** | **9.78** | **5.1** |

### F.2. Gate–Motion Consistency Analysis

This analysis evaluates whether the learned confidence gate correlates with a motion proxy and whether gate strength is predictive of temporal error trends, providing diagnostic cues for the role of gating in suppressing spurious motion responses before they affect temporal stability. To address this, we provide a targeted diagnostic that examines the relationship between the learned gate values, ground-truth motion magnitude, and temporal prediction error.

The analysis is conducted on held-out HuPR test sequences, where temporally aligned 3D pose annotations are available. For each frame $t$, we compute a ground-truth motion proxy

$$v_t = \frac{1}{J} \sum_{k=1}^{J} \|\mathbf{p}_{t+1,k} - \mathbf{p}_{t,k}\|_2,$$

and aggregate the learned gate values $g_{t,j}$ by averaging them within the corresponding spatial neighborhood, yielding a frame-level gate score $\bar{g}_t$.

**Correlation with Ground-Truth Motion**   Fig 6(a) shows the relationship between $\bar{g}_t$ and the ground-truth motion magnitude $v_t$. We observe a strong positive correlation (Pearson $r=0.72$), indicating that higher gate activations consistently correspond to frames with larger true motion. This result indicates that the learned gate is aligned with a motion-magnitude proxy on HuPR; however, it does not by itself distinguish human motion from clutter-induced Doppler responses, nor does it imply that $g_{t,j}$ is a calibrated estimate of signal quality.

**Error Stratification by Gate Strength**   To assess whether the gate selectively improves temporal accuracy when motion is present, we partition frames into quantiles based on $\bar{g}_t$ and compute MPJVE within each bin. As shown in Fig. 6(b), the ungated baseline exhibits steadily increasing MPJVE as gate quantiles rise, reflecting the greater difficulty of high-motion frames. In contrast, our gated model

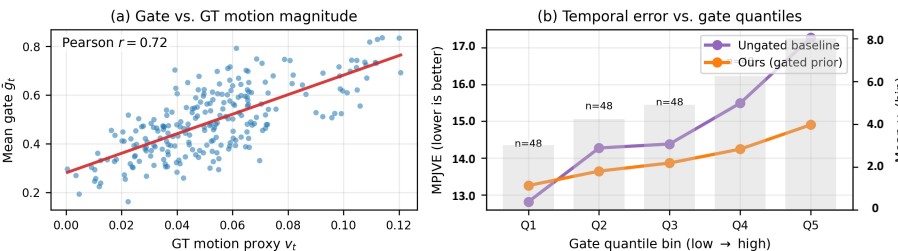

*Figure 6.* Interpretability check of the motion gate $g_{t,j}$ under single-frame evaluation on HuPR: correlation with GT velocity magnitude and binned MPJVE trends.

consistently achieves lower MPJVE across all bins, with the largest relative gains appearing in high-gate (high-motion) regimes. This pattern indicates that the motion-conditioned prior contributes most when motion cues are strong and relevant, rather than uniformly smoothing predictions.

Together, these diagnostics provide cues that the learned gate aligns with a motion-magnitude proxy and that its contribution is motion-dependent. While this analysis does not claim physical calibration of $g_{t,j}$ or explicit clutter rejection, it supports interpreting $g_{t,j}$ as an adaptive, pose-supervised motion-relevance prior rather than an incidental architectural artifact.

### F.3. Sensitivity to Gate Strength $\beta$

The gate-strength $\beta$ (Eq. 8) controls how strongly the learned motion relevance $g_{t,j}$ biases the cross-attention weights. To assess sensitivity, we vary $\beta$ while keeping all other components and training protocols unchanged, and evaluate on HuPR. Table 11 reports the results.

*Table 11.* Sensitivity to gate-strength $\beta$ under single-frame evaluation on HuPR.

| $\beta$ | MPJPE ↓ | MPJVE ↓ | AKV ↓ |
|---|---|---|---|
| 0 | 60.9 | 10.6 | 5.4 |
| 0.5 | 60.7 | 10.0 | 5.2 |
| 1 | 60.57 | 9.78 | 5.1 |
| 2 | 60.8 | 9.9 | 5.0 |
| 4 | 61.4 | 10.4 | 4.8 |

Overall, the performance is relatively insensitive within a moderate range of $\beta$, while extreme values may either weaken motion prompting ($\beta \to 0$) or over-amplify the gate bias (large $\beta$).

### F.4. Effect of Explicit Gate Supervision

Our motion gate $g_{t,j}$ is learned implicitly from pose supervision, without explicit gate-level labels. To further test whether direct supervision on the gate is necessary, we consider an auxiliary training-only objective that regresses a frame-level gate score $\bar{g}_t$ (defined in Appendix F.2) to the

ground-truth motion proxy $v_t$ (Eq. 4):

$$\mathcal{L}_{\text{gate}} = \|\bar{g}_t - \text{norm}(v_t)\|_2^2, \tag{21}$$

and add it to the total loss with a weight $\gamma$. Table 12 summarizes the comparison on HuPR.

*Table 12.* Ablation on explicit gate supervision (single-frame evaluation on HuPR).

| Setting | Gate–motion corr. $r$ ↑ | MPJPE ↓ | MPJVE ↓ | AKV ↓ |
|---|---|---|---|---|
| Implicit (ours $\gamma=0$) | 0.72 | 60.57 | 9.78 | 5.1 |
| Aux gate loss ($\gamma=0.1$) | 0.75 | 61.1 | 9.9 | 5.0 |

Results show that PULSE does not rely on explicit gate supervision: the gate already aligns with motion (Appendix F.2), and additional supervision is not necessary for temporal gains.

### F.5. Sensitivity to Cross-Attention Neighborhood Size

To examine the effect of neighborhood size, we vary it to aggregate Doppler tokens for each spatial token during conditional cross-attention, while keeping all other components unchanged. All models are trained and evaluated on the HuPR test split under identical settings in the single-frame setting.

Table 13 reports MPJPE and MPJVE under different neighborhood sizes. Two complementary trends can be observed. First, extremely local neighborhoods ($2 \times 2$) yield the lowest MPJPE but noticeably higher MPJVE, indicating that overly localized motion cues are insufficient to capture coherent velocity patterns. Second, as the neighborhood expands, MPJVE consistently decreases, suggesting improved access to motion-related Doppler cues. However, this comes at the cost of degraded MPJPE for larger neighborhoods (e.g., $7 \times 7$), reflecting reduced spatial specificity and increased interference from unrelated regions.

These results indicate that effective motion guidance requires a balance between locality and contextual coverage. Neither strictly local nor overly global aggregation is optimal: moderate neighborhoods (e.g., $3 \times 3$ or $5 \times 5$) achieve a better balance between spatial accuracy and temporal consistency.

*Table 13.* Effect of cross-attention neighborhood size on pose accuracy under single-frame evaluation (HuPR test set).

| Neighborhood size | MPJPE $\downarrow$ | MPJVE $\downarrow$ |
|:---:|:---:|:---:|
| 2×2 | 60.39 | 13.12 |
| 3×3 | 60.57 | 9.78 |
| 5×5 | 63.77 | 9.66 |
| 7×7 | 68.84 | 9.51 |

## F.6. Sensitivity to Spatial Patch Size

We further analyze the sensitivity to the spatial patch size used for tokenizing the range–angle magnitude map. Patch size governs the granularity of spatial representation and implicitly controls how much local structure is preserved before cross-domain interaction.

Table 14 summarizes performance under different patch sizes. Very small patches (2×2) exhibit substantially higher MPJPE, suggesting that overly fine-grained tokenization fragments spatial structure and weakens pose localization. At the other extreme, large patches (8×8) achieve competitive MPJPE but incur a sharp increase in MPJVE, indicating that excessive spatial aggregation suppresses localized motion cues and leads to temporally inconsistent predictions.

Intermediate patch sizes (4×4 and 6×6) strike a more favorable balance. In particular, 4×4 patches achieve strong spatial accuracy while maintaining low MPJVE, whereas 6×6 patches slightly favor spatial compactness at the expense of temporal precision. Overall, the results suggest that preserving sufficient spatial resolution is critical for motion-aware conditioning, while excessive aggregation—either too early or too coarse—diminishes the contribution of Doppler-derived priors.

*Table 14.* Effect of spatial patch size on pose accuracy under single-frame evaluation (HuPR test set).

| Patch size ($P_r \times P_a$) | MPJPE $\downarrow$ | MPJVE $\downarrow$ |
|:---:|:---:|:---:|
| 2×2 | 69.14 | 9.77 |
| 4×4 | 60.57 | 9.78 |
| 6×6 | 59.31 | 11.59 |
| 8×8 | 59.07 | 19.61 |

## F.7. Per-Joint Breakdown on HuPR

Table 15 reports a per-joint comparison on HuPR. AP follows the shared benchmark reported by milliMamba. MPJPE and MPJVE for milliMamba are measured from our reproduction because the original paper reports AP metrics only. The largest MPJVE gains of PULSE appear at wrists and ankles, which are high-velocity distal joints with stronger Doppler signatures and higher sensitivity to clutter-driven contamination.

*Table 15.* Per-joint AP, MPJPE, and MPJVE on HuPR. milliMamba MPJPE and MPJVE are measured from our reproduction.

| Method | Metric | Head | Neck | Sho. | Elbow | Wrist | Hip | Knee | Ankle |
|--------|--------|------|------|------|-------|-------|-----|------|-------|
| milliMamba | AP | 90.0 | 91.8 | 83.2 | 75.2 | 59.5 | 94.3 | 93.6 | 89.3 |
| milliMamba | MPJPE | 53.1 | 48.7 | 58.2 | 69.5 | 90.2 | 45.0 | 62.3 | 78.0 |
| milliMamba | MPJVE | 8.2 | 7.5 | 9.4 | 11.1 | 14.5 | 7.2 | 10.2 | 13.8 |
| PULSE (1F) | AP | 91.8 | 93.2 | 84.9 | 77.8 | 62.4 | 95.7 | 94.6 | 90.5 |
| PULSE (1F) | MPJPE | 51.3 | 47.2 | 56.4 | 67.1 | 87.4 | 43.2 | 60.1 | 76.3 |
| PULSE (1F) | MPJVE | 7.8 | 7.1 | 8.9 | 10.4 | 13.8 | 6.9 | 9.7 | 13.1 |

