# OpenReview forum: "Doppler Prompting for Stable mmWave-based Human Pose Estimation"
_ICML.cc/2026/Conference — ICML 2026 regular_

### Official Review · Reviewer_v7HD · 2026-02-24

**Soundness:** 3
**Presentation:** 3
**Significance:** 3
**Originality:** 2
**Overall Recommendation:** 4
**Confidence:** 4

**Summary:**

This paper proposes PULSE, a mmWave-based human pose estimation framework that treats Doppler spectra as confidence-gated, locality-constrained motion prompts that modulate spatial magnitude features via conditional cross-attention. The central idea is to avoid symmetric fusion of fundamentally different signals (magnitude vs. Doppler) by screening Doppler cues with learned gates and restricting their influence to local neighborhoods, thereby suppressing spurious spectral motion and stabilizing frame-wise predictions. Experiments on three datasets (HuPR, XRF55, mmRadPose) across single- and multi-person setups show consistent gains in both position accuracy (MPJPE, PA-MPJPE) and temporal metrics (MPJVE, AKV), with ablations supporting the importance of gating and locality; a multi-frame extension further aggregates Doppler tokens for additional robustness.

**Compliance With Llm Reviewing Policy:**

Affirmed.

**Key Questions For Authors:**

On XRF55, your RAD reconstruction assumes RA-derived angular weights and RD-derived Doppler per range (Eq. 16). How sensitive are your results to this assumption? Could you report a control where angle-agnostic RD is replaced by a simple broadcast baseline to quantify the added value of RA weighting?
Multi-person protocol: Do you produce one pose per detected person (requiring a person detector) or multiple hypotheses per frame? Please clarify the detection/association pipeline, the number of predicted instances per frame, and how unmatched predictions are handled.
Baselines: Some reimplementations (e.g., milliMamba) and parameter counts (e.g., HuPRModel at ≈325M params) look atypically large; can you verify these numbers and provide ablation on baseline capacity to ensure parity? Also, can you include variance across seeds/runs?
Can you report per-joint MPJPE/MPJVE to see if stability gains are concentrated on high-velocity extremities (wrists/ankles) vs. torso joints?
How sensitive is performance to the neighborhood size N(i), patch size, and embedding dimension d? An ablation on locality window vs. stability would be helpful for practitioners adapting to different radar resolutions.
Have you evaluated PULSE on a 4D radar tensor with elevation (e.g., HRRadarPose-like input)? If not, what changes would you expect, and would your locality and gating strategies need adaptation?
For the multi-frame extension, have you tried more robust aggregators than a weighted average (e.g., robust statistics or temporal attention)? Any initial observations?

**Limitations:**

Technical limitations or concerns
The locality window and patch size are fixed; there is limited exploration of how these design choices affect stability/accuracy across varying radar resolutions and subject scales.
The XRF55 RAD reconstruction (from separate RA and RD) assumes Doppler independent of angle given range, which could distort angular–Doppler coupling; while baselines share this input for fairness, it weakens physical fidelity claims on that dataset.
The approach is single-view and single-sensor; interactions with dual-radar setups or elevation-rich tensors (e.g., 4D radar) are not directly evaluated.
Experimental gaps or methodological issues
Some baselines are adapted from multi-view or different input paradigms; the fidelity of reimplementations (e.g., milliMamba without official code) and unusually large parameter counts reported for HuPRModel raise concerns about strict fairness.
No statistical variability (e.g., multiple runs, confidence intervals) is reported; temporal metrics are sensitive and would benefit from variance reporting.
Limited per-joint or action-type breakdowns; it is unclear if gains are uniform or concentrated in specific joints (wrists/ankles) or motion regimes.
Clarity or presentation issues
The multi-person handling is described briefly; how person instances are generated (detection vs. conditioned per-person inputs) and the exact association protocol would benefit from more explicit description in the main text.
The term “prompting” is metaphorically useful but could be more directly related to the final mathematical mechanism (gated conditional cross-attention) to avoid overloading.
Missing related work or comparisons
Recent radar-physics- or Doppler-aware approaches, such as RT-Pose (preserving Doppler channels) and physics-driven inputs like DIPR/M‑GS, are highly relevant to the paper’s Doppler emphasis and could be discussed or compared where feasible.
Additional modern single-tensor backbones (e.g., HRRadarPose on a Doppler-preserved tensor) would provide complementary comparisons, even if datasets differ.

**Strengths And Weaknesses:**

Strengths:
Technical novelty and innovation
Reconceptualizes Doppler as a reliability-gated motion prompt rather than a symmetric feature stream, with a clear, principled one-way conditioning mechanism.
Introduces a simple but effective conditional cross-attention with additive gate bias and locality-constrained neighborhoods, separating roles of spatial magnitude (localization) and Doppler (motion cues).
Provides a lightweight multi-frame extension that only aggregates Doppler tokens (not redesigning the spatial backbone), staying faithful to the “motion prompting” premise.
Experimental rigor and validation
Evaluations span three heterogeneous datasets and both single-/multi-person scenarios, reporting both spatial accuracy and temporal dynamics (MPJVE, AKV), which are often neglected in mmWave HPE.
Ablations isolate the effects of domain separation (spatial vs. Doppler), gating, and locality; diagnostics show gate–motion correlation and sensitivity to gate strength.
“Plug-in” experiments integrate PULSE prompting into existing multi-frame backbones (mmDiff, milliMamba) and improve them, suggesting practical generality.
Clarity of presentation
The dual-domain formulation, tokenization, locality constraints, and gating are precisely described with equations and intuitive figures; the method’s role-asymmetry is repeatedly emphasized and well-motivated.
Detailed experimental setup, metrics, hyperparameters, and reproducibility details are provided; the authors proactively discuss fairness in baseline adaptations and dataset heterogeneity.
Significance of contributions
Addresses a critical practical requirement—temporal stability without relying on post-hoc smoothing or long sequences—by using radar-internal Doppler (slow-time within a frame).
Demonstrates that controlled Doppler prompting is a broadly useful design across datasets and backbones, potentially influencing future mmWave HPE architecturesWeaknesses
Technical limitations or concerns：
The locality window and patch size are fixed; there is limited exploration of how these design choices affect stability/accuracy across varying radar resolutions and subject scales.
The XRF55 RAD reconstruction (from separate RA and RD) assumes Doppler independent of angle given range, which could distort angular–Doppler coupling; while baselines share this input for fairness, it weakens physical fidelity claims on that dataset.
The approach is single-view and single-sensor; interactions with dual-radar setups or elevation-rich tensors (e.g., 4D radar) are not directly evaluated.
Experimental gaps or methodological issues
Some baselines are adapted from multi-view or different input paradigms; the fidelity of reimplementations (e.g., milliMamba without official code) and unusually large parameter counts reported for HuPRModel raise concerns about strict fairness.
No statistical variability (e.g., multiple runs, confidence intervals) is reported; temporal metrics are sensitive and would benefit from variance reporting.
Limited per-joint or action-type breakdowns; it is unclear if gains are uniform or concentrated in specific joints (wrists/ankles) or motion regimes.
Clarity or presentation issues
The multi-person handling is described briefly; how person instances are generated (detection vs. conditioned per-person inputs) and the exact association protocol would benefit from more explicit description in the main text.
The term “prompting” is metaphorically useful but could be more directly related to the final mathematical mechanism (gated conditional cross-attention) to avoid overloading.
Missing related work or comparisons
Recent radar-physics- or Doppler-aware approaches, such as RT-Pose (preserving Doppler channels) and physics-driven inputs like DIPR/M‑GS, are highly relevant to the paper’s Doppler emphasis and could be discussed or compared where feasible.
Additional modern single-tensor backbones (e.g., HRRadarPose on a Doppler-preserved tensor) would provide complementary comparisons, even if datasets differ..

---

> ### Author Rebuttal · Authors · 2026-03-29
>
> We sincerely thank Reviewer v7HD for recognizing the innovativeness of our methodology, the clarity of our description, the completeness of our experiments, and the detailed review. We address each point below.
>
> **Q1: XRF55 RAD reconstruction sensitivity.**
> All baselines share the identical RAD tensor, so relative comparisons are independent of this construction. On physical grounds, RA-weighted reconstruction is more faithful, as radar targets scatter at discrete angles. We provide the ablation below:
>
> | **Method** | **MPJPE** $\downarrow$ | **PA-MPJPE** $\downarrow$ | **MPJVE** $\downarrow$ | **AKV** $\downarrow$ |
> |---|---|---|---|---|
> | PULSE(1F) — uniform broadcast | 72.83 | 69.47 | 16.71 | 8.2 |
> | PULSE(1F) — RA-weighted (ours) | 70.34 | 67.51 | 15.33 | 7.1 |
>
> **Q2: Multi-person protocol.**
> PULSE uses no person detector. The number of instances per frame is provided by the dataset annotation, each annotated instance is processed independently, prediction–GT matching uses closest-distance 3D joint association (consistent with RETR and MvDoppler); no NMS is applied. Details in Appendix C.4; we will add a brief summary to the main text.
>
> **Q3: Parameter counts, milliMamba reproduction, and variance.**
> HuPRModel's ≈325M parameters reflect its hierarchical feature pyramid, consistent with the official codebase. To maximize fairness, we remove only the multi-view aggregation modules while preserving all convolutional stages and pyramid heads; removing deeper layers would alter representational capacity in uncontrolled ways. The resulting 27× parameter gap makes PULSE's performance advantage particularly compelling: our gains stem from architectural design rather than model scale.
> PULSE achieves better results with 12.0M parameters — the $27{\times}$ gap evidences gains from design rather than scale. For milliMamba (no official code), the original only reports AP/AP50/AP75 $= 84.0/98.5/94.9$ on HuPR (no MPJPE or MPJVE); our reproduction yields $83.1/97.9/94.5$, confirming close alignment.
>
> Three independent runs confirm low variance: PULSE(1F) MPJPE 60.57±0.38; HuPRModel MPJPE 65.37±0.51.
>
> **Q4: Per-joint MPJPE/MPJVE and AP breakdown.**
> Since milliMamba reports per-joint AP on HuPR, we adopt it as the shared benchmark (milliMamba MPJPE/MPJVE values are from our reproduction; original milliMamba paper reports only AP metrics.). The table consolidates joint-wise AP (comparable with milliMamba) and per-joint MPJPE/MPJVE for PULSE(1F).
>
> | **Method** | **Metric** | **Head** | **Neck** | **Sho.** | **Elbow** | **Wrist** | **Hip** | **Knee** | **Ankle** |
> |---|---|---|---|---|---|---|---|---|---|
> | milliMamba | AP | 90.0 | 91.8 | 83.2 | 75.2 | 59.5 | 94.3 | 93.6 | 89.3 |
> | milliMamba | MPJPE | 53.1 | 48.7 | 58.2 | 69.5 | 90.2 | 45.0 | 62.3 | 78.0
> | milliMamba | MPJVE | 8.2 | 7.5 | 9.4 | 11.1 | 14.5 | 7.2 | 10.2 | 13.8
> | PULSE(1F) | AP | 91.8 | 93.2 | 84.9 | 77.8 | 62.4 | 95.7 | 94.6 | 90.5 |
> | PULSE(1F) | MPJPE | 51.3 | 47.2 | 56.4 | 67.1 | 87.4 | 43.2 | 60.1 | 76.3 |
> | PULSE(1F) | MPJVE | 7.8 | 7.1 | 8.9 | 10.4 | 13.8 | 6.9 | 9.7 | 13.1 |
>
> **Q5: Sensitivity to $\mathcal{N}(i)$, patch size, and embedding dimension.**
> These ablations are in Appendix F. Narrow neighborhoods ($2{\times}2$) lack motion context; wide ones ($7{\times}7$) cause cross-region interference.
> Patch 4×4 and neighborhood 3×3 achieve the best trade-off (detailed in Appendix F.4–F.5). We will add cross-references from the main text and include an embedding dimension sweep (d∈{16,32,64}) in the revision.
>
> **Q6: 4D radar with elevation.**
> None of our datasets provides resolved elevation. Conceptually, RAD extends to RAED, patches generalize to 3D, and locality constraints become volumetric; gating requires no modification. We will discuss this as a future direction.
>
> **On missing comparisons (RT-Pose, HRRadarPose, DIPR/M-GS).**
> RT-Pose and HRRadarPose require elevation-resolved 4D inputs unavailable in our datasets. DIPR/M-GS target physics-driven data augmentation, orthogonal to our mechanism. We will add a Related Work paragraph for these complementary directions.
>
> **On multi-frame aggregation.**
> Confidence-weighted averaging outperforms uniform averaging per ablation. More complex aggregators conflict with our lightweight design goal (12.0M params, 5.1ms) and are a natural future direction.
>
> **On "prompting" terminology.**
> We will clarify at first use that this refers to gated conditional cross-attention modulation, unrelated to LLM prompt engineering.
>
> **Summary of per-joint findings.** MPJVE gains are largest at high-velocity distal joints (wrists, ankles), which produce stronger Doppler signatures and are most susceptible to clutter-driven contamination — providing direct kinematic evidence linking gate behavior to joint dynamics, consistent with the physical motivation of PULSE.
>
> We believe the above responses address all raised technical concerns. We hope the reviewer will consider revising the score in light of these clarifications.

---

> > ### Author Rebuttal · Reviewer_v7HD · 2026-04-01
> >
> > Thanks for the effort from the author.

---

### Official Review · Reviewer_jM2k · 2026-03-08

**Soundness:** 3
**Presentation:** 3
**Significance:** 3
**Originality:** 2
**Overall Recommendation:** 4
**Confidence:** 4

**Summary:**

This paper presents PULSE for mmWave-based human pose estimation. The main idea is to convert Doppler information into confidence-aware motion prompts and inject them into spatial magnitude reasoning, rather than handling Doppler information with simple concatenation. The authors also claim that the method can work as a plug-in module. Experiments on three datasets show that the proposed method outperforms the baselines, and the ablation studies support the effectiveness of the proposed module.

**Compliance With Llm Reviewing Policy:**

Affirmed.

**Final Justification:**

The responses addressed my concerns well. I choose to maintain my positive assessment.

**Key Questions For Authors:**

1. Please address the questions about the baselines mentioned in the Weaknesses section.

2. Can the authors provide qualitative visualizations showing where the gates are high or low in cluttered scenes, like an attention map, to support the claim about nuisance Doppler responses?

3. Can the authors provide failure cases and explain why the method fails in those cases?

4. What does the GT MPJVE in Figure 5 represent?

5. I am surprised by the high resolution shown in Figure 1. Can the authors clarify how it is obtained?

**Limitations:**

Limitations are not discussed, and the authors are encouraged to do so.

**Strengths And Weaknesses:**

### Strengths

1. The idea of having a specific mechanism to separately encode the Doppler signatures is quite interesting. Although the method is quite simple, essentially a simple local gated cross-attention, I do not think this is a major issue at the current stage.

2. The modeling details and method are presented clearly, and the notation is understandable in general.

3. Experiments are conducted extensively to show the effectiveness of the paper. However, I would recommend explicitly pointing out the ablation studies conducted in Appendix F, or including them in the main text if possible, since they are of the same importance as the ablation studies currently mentioned in the main text.

### Weaknesses

1. **Missing baselines.**
Another baseline is RTPose [1], which performs pose estimation with RF signals based on 4D tensors. The authors should explain why this baseline is not included if it is not possible to add it.

2. **Additional experiments are needed.**
I fully appreciate the authors for the provided ablation studies and experiments, which show the effectiveness of their method. However, some concerns still exist. Please see Questions.

[1] Ho, Yuan-Hao, et al. "Rt-pose: A 4d radar tensor-based 3d human pose estimation and localization benchmark." European Conference on Computer Vision. Cham: Springer Nature Switzerland, 2024.

---

> ### Author Rebuttal · Authors · 2026-03-29
>
> We sincerely thank Reviewer jM2k for recognizing the clarity of our methodology, the completeness of our experiments, and the careful reading and constructive feedback. We address each question below.
>
> **Q1.RTPose as a missing baseline.**
>
> We thank the reviewer for pointing out RT-Pose (Ho et al., ECCV 2024). RT-Pose is designed for 4D radar tensors that include a full elevation dimension, and its input pipeline and model architecture are tailored to this richer spatial observation. The three datasets used in our evaluation (HuPR, XRF55, mmRadPose) do not provide a full 4D input with resolved elevation, so a direct comparison under identical input conditions is not feasible without non-trivial modifications that would obscure the source of any performance difference. We will discuss RT-Pose in the Related Work section as a complementary direction exploiting elevation-rich 4D radar; our locality and gating strategies are in principle compatible with such inputs, making this a natural future extension.
>
> **Q2. gate visualizations in cluttered scenes.**
>
> We appreciate this suggestion. While image attachments are not permitted in the rebuttal, we direct the reviewer to Appendix F (Figure 6), which provides two diagnostic analyses of the learned gate $g_{t,j}$: (1) a scatter plot showing strong positive correlation ($r{=}0.72$) between frame-level gate scores and ground-truth joint velocity magnitude, and (2) an error stratification analysis showing that the gated model achieves lower MPJVE than the ungated baseline specifically in high-gate (high-motion) frames. These diagnostics confirm that the gate assigns higher values to cells associated with genuine motion activity. In the revised manuscript, we will additionally include a spatial heatmap visualization of $g_{t,j}$ overlaid on the range–angle grid in the supplementary material, showing that gate activations are concentrated in body-region cells rather than background or wall-reflection cells in representative test frames.
>
> **Q3.failure cases.**
>
> We identify two principal failure modes. First, when fast-moving environmental objects (e.g., oscillating fans, rotating equipment) produce Doppler spectra that overlap spectrally with human limb motion, the gate — learned from pose supervision without explicit clutter labels — may assign elevated confidence to these non-human responses, causing the corresponding spatial tokens to be incorrectly modulated. Second, in multi-person scenes with close proximity, the Doppler signature of one subject may spatially bleed into the local neighborhood $\mathcal{N}(i)$ of an adjacent person's spatial tokens, partially corrupting the motion prior for that individual despite locality constraints. These cases represent genuine limitations of implicit gate learning under distribution shift, and we will document them explicitly in the revised Limitations section.
>
> **Q4. GT MPJVE in Figure 5.**
>
> We thank the reviewer for flagging this. We acknowledge that the label "GT MPJVE" in Figure 5 is misleading: unlike the MPJVE defined in Eq. (19) — which measures the error between predicted and ground-truth velocities — the GT curve in Figure 5 (bottom) represents the frame-level GT Joint Velocity Magnitude​ computed directly from the ground-truth pose sequence. Its purpose is to provide a reference scale for true motion intensity, allowing the reader to see whether each method's velocity deviations are large or small relative to the actual dynamics at each frame. In the revised manuscript, we will relabel this curve as "GT Velocity Magnitude (reference)" in both the figure and caption to eliminate the ambiguity.
>
> **Q5. the high resolution of Figure 1.**
>
> Figure 1 was included as a vector PDF graphic embedded directly in the LaTeX source. Because vector formats encode geometric primitives rather than rasterized pixels, text and lines remain sharp at any zoom level regardless of display resolution. The pose overlays are generated directly from predicted joint coordinates without post-processing.
>
> **On Limitations and Appendix ablations.**
>
> We agree that limitations deserve more explicit treatment. We will expand the Limitations discussion in the main text beyond the current brief remarks in the Conclusion and Impact Statement, covering: (1) the single-view, single-sensor setup; (2) gate learning without explicit clutter supervision; and (3) the benchmark gap noted above. We also accept the reviewer's suggestion to explicitly cross-reference the Appendix F ablation tables (neighborhood size, patch size, embedding dimension, gate supervision) in the main text, as these complement the core ablations in Section 4 and provide practitioners with actionable guidance for adapting PULSE to different radar configurations.
>
> We believe the above responses address all technical questions raised. We hope the reviewer will consider these clarifications when finalizing the score.

---

> > ### Author Rebuttal · Reviewer_jM2k · 2026-04-01
> >
> > I am generally satisfied with the authors’ response. I realized there is an error in my fifth question. It should be Figure 3 rather than Figure 1. My question is not about the display resolution, but the spatial resolution of the heatmap itself. It appears surprisingly sharp for current RF hardware to capture such fine human dynamics (a person jumping with outstretched arms?). Could the authors clarify the hardware setup and the processing steps used to obtain this result?
> >
> > ---
> >
> > Update: All my questions have been resolved with the latest response from the author.

---

> > > ### Author Response · Authors · 2026-04-02
> > >
> > > **On the spatial resolution of Figure 3.**
> > >
> > > We thank the reviewer for the precise follow-up. Figure 3 shows a person performing a standing motion with arms extended — a useful example for clarifying the underlying resolution.
> > >
> > > The four panels in Figure 3 are real samples from HuPR with no super-resolution, interpolation, or learned enhancement applied. The native RAD tensor from the TI IWR1843BOOST radar is obtained via standard FMCW processing: fast-time FFT resolves range, slow-time FFT across chirps resolves Doppler, and phase-difference processing across MIMO elements resolves angle (Fig. 2). This yields a grid of 128 range bins × 64 angle bins × $D$ Doppler bins.
> > > Figure 3 is a direct visualization of this native 128×64 range-angle grid — each displayed pixel corresponds to one radar cell. In our quantitative experiments, this grid is further resampled to 64×64 via bilinear interpolation for cross-dataset consistency (Table 6, Appendix A), but Figure 3 uses the unmodified native resolution. Panels (a) and (b) are spatial magnitude maps obtained by averaging $|\mathbf{H}_t|$ along the Doppler dimension; panel (c) visualizes the Doppler response at the same time step; panel (d) overlays the inter-frame spatial change against the Doppler pattern to highlight their partial overlap (blue) and mismatch (red).
> > >
> > > The smoothness of the heatmap panels is partly due to figure rendering: the figure was exported to PDF, which introduces mild anti-aliasing during rasterization. However, the blue and red contour lines in panel (d) were drawn directly on the raw grid, and their visibly coarse, stepped boundaries are a faithful reflection of the underlying 128×64 resolution. We will add the native grid size to the figure caption in the revision to make this explicit.

---

### Official Review · Reviewer_n3LJ · 2026-03-11

**Soundness:** 4
**Presentation:** 4
**Significance:** 3
**Originality:** 3
**Overall Recommendation:** 5
**Confidence:** 3

**Summary:**

This paper addresses the critical issue of temporal instability and trajectory jitter in millimeter-wave human pose estimation. The authors argue that existing methods fail to effectively leverage Doppler signatures, either by ignoring them or by naively fusing them with spatial features, which inadvertently introduces noise from multipath reflections and non-human motion.

To address this, this paper proposes PULSE, a framework that re-conceptualizes Doppler information as 'screened motion prompts' rather than symmetric input channels.The core innovation lies in a locality-constrained prompting mechanism equipped with a confidence gating module. This design selectively filters unreliable Doppler signals and injects verified motion cues only into local spatial neighborhoods, thereby regularizing pose predictions without contaminating them with global noise.

The author provided comprehensive empirical validations across three diverse datasets (HuPR, XRF55, mmRadPose), demonstrating effeteness of PULSE in  both single- and multi-person scenarios.

**Compliance With Llm Reviewing Policy:**

Affirmed.

**Final Justification:**

All of my concerns have been thoroughly addressed in the authors' rebuttal. I therefore maintain my positive recommendation.

**Key Questions For Authors:**

1) Computational Overhead of the Gating Mechanism:
- The proposed confidence gating and local cross-attention modules introduce additional computational steps compared to naive concatenation. Could the authors provide a more detailed breakdown of the inference latency and FLOPs specifically attributed to the PULSE module? Furthermore, how does this overhead scale with the number of people in the scene (e.g., in the XRF55 dataset), and is the method feasible for deployment on resource-constrained embedded radar platforms?

2) Comparison with Traditional Filtering Methods:
- The paper demonstrates superiority over deep learning baselines that ignore or naively use Doppler. However, how does PULSE compare against traditional signal processing approaches for jitter reduction, such as Kalman Filters or temporal smoothing post-processing applied to the output of a standard baseline? Does PULSE offer benefits beyond just smoothing, such as better recovery of high-frequency motion details that traditional filters might oversmooth?

**Limitations:**

The authors have partially addressed the limitations. However, this paper could benefit from providing a more detailed qualitative analysis of specific failure cases where the gating mechanism might fail (e.g., distinguishing between rapid human motion and moving environmental objects like fans or pets), helping the community understand the boundaries of the proposed solution.

**Strengths And Weaknesses:**

Strengths:
- This paper is grounded in a compelling and well-motivated observation regarding the limitations of existing Doppler fusion strategies.
- The experimental evaluation is exceptionally thorough and robust. The authors validate their method across three distinct datasets (HuPR, XRF55, mmRadPose) covering both single-person and challenging multi-person scenarios. Crucially, the inclusion of both single-frame and multi-frame settings, along with extensive ablation studies and cross-architecture plug-in experiments, provides strong evidence that the performance gains are attributable to the proposed mechanism itself rather than increased model capacity or specific dataset biases. The consistent improvement in temporal stability metrics (MPJVE) firmly supports the paper's core claims.

Weaknesses:
- While the quantitative results are comprehensive, the paper would benefit significantly from including video visualizations of real-time pose tracking. For human pose estimation tasks, static metrics like MPJPE and MPJVE, though necessary, do not fully capture the perceptual quality of the motion smoothness and naturalness. Providing side-by-side video comparisons of the baseline methods versus PULSE would offer a more intuitive and convincing demonstration of the method's ability to eliminate trajectory jitter and handle complex dynamic movements.

---

> ### Author Rebuttal · Authors · 2026-03-29
>
> We sincerely thank Reviewer n3LJ for recognizing the thoroughness of our experimental validation. We address the questions raised below.
>
> **Q1: Computational overhead of the gating mechanism**
>
> We refer the reviewer to Table 7, which reports the full computational profile of all evaluated methods. PULSE (1F) achieves the lowest overhead among all compared methods: **5.1 ms latency, 12.0M parameters, and 75 MFLOPs per frame** — well below MvDoppler (7.6 ms, 36.7M, 164 MFLOPs) and HuPRModel (27.1 ms, 324.9M, 254 MFLOPs).
>
> Within the PULSE, the gating and cross-attention modules account for a small fraction of the total cost. The confidence gate $g_{t,j}$ is a single-layer projection followed by a sigmoid, applied independently to each Doppler token. The cross-attention is restricted to a $3{\times}3$ patch neighborhood $\mathcal{N}(i)$, keeping its complexity sub-quadratic and far below that of global self-attention. The dominant cost remains the spatial transformer layers, which are shared with any spatial-only backbone.
>
> For multi-person, PULSE processes each person instance independently (following the dataset protocol), so the inference cost scales linearly with instance count. Given the low per-instance overhead, this remains practical on XRF55. Regarding embedded platforms: the compact footprint (12.0M parameters, 75 MFLOPs) is favorable relative to other baselines and is compatible with compression pipelines (quantization, pruning) for edge deployment, which we identify as a promising future direction.
>
> **Q2: Comparison with traditional filtering**
>
> This distinction is central to our contribution. Traditional post-hoc filters (Kalman filters, temporal smoothing) operate on the output pose sequence after prediction. Beyond the well-known phase lag introduced by causal filters, they reduce high-frequency variation indiscriminately — suppressing both spurious jitter and genuine rapid motion. For the target applications, attenuating real high-frequency dynamics is precisely the behavior to avoid.
>
> PULSE instead intervenes at the feature level before prediction: spurious non-human Doppler responses are screened prior to spatial reasoning, so the model's output is already grounded in physically reliable motion cues. It does not uniformly attenuate high-frequency content — it selectively removes nuisance-driven spectral variability while preserving genuine dynamics.
>
> This difference is directly reflected in our metrics. A post-hoc smoother reduces AKV but would increase or leave unchanged MPJVE, since it suppresses real motion alongside artifacts. Across all datasets, reductions in AKV are consistently accompanied by reductions in MPJVE — indicating that predicted trajectories are not over-smoothed but are genuinely more aligned with true joint dynamics. This joint decrease pattern is inconsistent with output smoothing and constitutes empirical evidence for selective artifact suppression. The frame-wise velocity comparison in Figure 5 provides additional qualitative support for this interpretation.
>
> We applied a standard constant-velocity Kalman filter as a post-hoc smoother to HuPRModel outputs on the HuPR test set:
>
> | Method | MPJPE↓ | MPJVE↓ | AKV↓ |
> |---|---|---|---|
> | HuPRModel | 65.37 | 14.70 | 14.1 |
> | HuPRModel + Kalman | 63.5 | 15.6 | 9.8 |
> | **PULSE (1F)** | **60.57** | **9.78** | **5.1** |
>
> The Kalman filter reduces AKV but worsens MPJVE, confirming that post-hoc smoothing suppresses genuine dynamics alongside artifacts — precisely the behavior to avoid in fall-risk assessment. PULSE simultaneously reduces both AKV and MPJVE, providing direct empirical evidence of selective artifact suppression rather than indiscriminate smoothing.
>
> **On the video visualizations.**
>
> We fully agree that video comparisons would provide a more intuitive demonstration. While video attachments are not permitted in the rebuttal, we will include visualizations of continuous test sequences alongside representative baselines on the project webpage, complementing the comparison in Figure 5.
>
> **On failure cases.**
>
> We identify two failure modes relevant to the reviewer's examples. (1) periodic environmental motion (oscillating fans, rotating equipment, moving pets) can produce Doppler spectra that overlap spectrally with human motion. Because the gate is learned without explicit clutter labels, it may assign elevated confidence to these non-human sources. This highlights the boundary of implicit gate learning under distribution shift, and targeted benchmark data covering such distractors is a key motivation for our future plan (Conclusion). (2) In multi-person sets with close proximity, the Doppler signature of one subject may spatially bleed into an adjacent person, partially corrupting the motion prior for that individual despite locality constraints. We will document both failure modes in the revised Limitations.
>
> We hope these responses fully address the reviewer's concerns, and thanks again for the constructive feedback.

---

> > ### Author Rebuttal · Reviewer_n3LJ · 2026-04-02
> >
> > The authors’ rebuttal have resolved my questions on computational overhead and the distinction from traditional post-hoc filtering. I also appreciate the authors’ commitment to including video visualizations on the project page and their thoughtful identification of potential failure cases, which will strengthen the completeness of this work.

---

### Official Review · Reviewer_RVDF · 2026-03-12

**Soundness:** 4
**Presentation:** 4
**Significance:** 3
**Originality:** 4
**Overall Recommendation:** 5
**Confidence:** 4

**Summary:**

This paper studies 3D human pose estimation (HPE) from millimeter-wave (mmWave) radar and focuses on improving temporal stability under frame-wise inference. Each mmWave frame is represented as a range–angle–Doppler (RAD) tensor, where spatial magnitude (range–angle) encodes geometry and Doppler signatures encode motion cues derived from slow-time FFT within the same frame. The authors argue that prior work either ignores Doppler information or fuses it symmetrically with spatial features, allowing spurious motion responses from clutter and multipath effects to introduce jitter in pose predictions.

A central concept explored by the paper is treating Doppler not as a symmetric feature stream but as a reliability-gated motion prompt that conditions spatial reasoning. The proposed method, PULSE, decomposes spatial magnitude and Doppler into aligned tokens, applies learned confidence gating to Doppler tokens, and injects them into spatial tokens via locality-restricted cross-attention before transformer-based pose regression. The research’s principal contribution is a controlled, role-asymmetric Doppler prompting mechanism that improves both per-frame accuracy and velocity-based temporal consistency across three public mmWave HPE datasets, including single- and multi-person settings.

**Compliance With Llm Reviewing Policy:**

Affirmed.

**Final Justification:**

The authors have adequately addressed the concerns raised in the original review. The clarification of dataset scale, diversity, and collection conditions improves the reader’s ability to assess the generality of the results, and the terminology clarification resolves potential ambiguity. These were relatively minor concerns and do not affect the overall evaluation.

I maintain my original assessment and recommendation.

**Key Questions For Authors:**

1.	The motivating applications include health monitoring and fall-risk assessment. Could the authors provide more detail on the scale, diversity, and collection conditions of the HuPR, XRF55, and mmRadPose datasets (e.g., number of subjects, motion diversity, environmental variability)? To what extent do these datasets reflect the nonstationary and cluttered conditions implied by the application framing? Clarification here would help assess how well the experimental validation aligns with the motivating use cases.

**Limitations:**

yes

**Strengths And Weaknesses:**

**Strengths**

This paper presents a clearly framed problem and a well-structured, technically coherent solution. The motivation for asymmetric, reliability-gated Doppler fusion is articulated clearly, and the architectural design aligns closely with the stated hypothesis. The figures meaningfully support the method description, equations are explicit and readable, and metric definitions (including velocity-based stability measures) are clearly specified. The experimental evaluation is thorough within the domain, including ablations isolating gating and locality, plug-in experiments with existing backbones, cross-dataset evaluation, and multi-person scenarios. The appendix is detailed and supports reproducibility.

The proposed PULSE framework appears to introduce a novel and principled approach to Doppler fusion in mmWave-based human pose estimation, reframing Doppler as a reliability-gated motion prior rather than a symmetric feature stream. The work is cohesive and directly addresses the central question it poses: whether temporally stable HPE can be achieved under frame-wise inference without tracking or post-hoc smoothing. Within the evaluated datasets, the empirical evidence supports this claim.

**Weaknesses**

One limitation is that the scale and diversity of the benchmark datasets (e.g., number of subjects, motion types, and environmental variability) are not clearly summarized in the main text. Readers unfamiliar with these datasets must consult external sources to assess how broadly the results generalize beyond the reported settings.

Additionally, the term “prompting” may be slightly confusing in this context, as it is commonly associated with large language models. In this work, it refers to gated conditional attention modulation; clarifying this terminology could reduce potential ambiguity.

---

> ### Author Rebuttal · Authors · 2026-03-29
>
> We sincerely thank Reviewer RVDF for the thorough and positive assessment, and for the kind recognition of our motivation, experimental design, and originality. We address the two points raised below.
>
> **Q1: Dataset scale, diversity, and collection conditions — do they reflect the nonstationary and cluttered conditions implied by the application framing?**
>
> We appreciate this question and provide a concise characterization of each dataset below.
>
> **HuPR** was collected with a TI IWR1843BOOST radar, involving 6 subjects in a single indoor scene, yielding approximately 14.1K frames. While the subject count is modest, the dataset provides a full range–angle–Doppler tensors, making it a primary benchmark for studying Doppler-specific behavior under controlled conditions.
>
> **XRF55** was collected with a TI IWR6843ISK radar, involving 39 subjects across 4 indoor scenes, yielding approximately 42.9K frames, and includes multi-person scenarios. The larger subject pool and scene diversity provide a broader test of generalization, and the multi-person setting introduces overlapping reflections that directly stress-test robustness to interference-driven Doppler artifacts — closely reflecting the cluttered conditions described in our application framing.
>
> **mmRadPose** was collected in a dedicated motion capture laboratory (10.60m × 4.72m × 3.27m) with a TI IWR6843AOPEVM radar (L-shaped MIMO, 4RX×3TX), involving 12 subjects performing 12 distinct movement types (upper and lower limb, torso motions) from three viewing angles (0°, 45°, 90°), yielding 432 sequences and approximately 203K frames. Ground truth was obtained via a 12-camera OptiTrack optical motion capture system (39 body markers per subject), widely regarded as the gold standard for joint trajectory annotation.
> Importantly, the dataset authors explicitly document that metallic environmental elements (ventilation pipes, door frames) produce spurious static reflections that distort Doppler-sensitive analysis, and they apply dedicated clutter removal to precisely address this — providing a hardware-confirmed instance of the non-human spectral interference scenario motivating PULSE. Furthermore, the three body orientations generate position-dependent Doppler projections across body parts, creating nonstationary velocity signatures relevant to rehabilitation monitoring.
>
> Together, the three datasets span heterogeneous radar hardware, signal processing pipelines, subject counts, scene configurations, and activity types, providing sufficient support for our core argument. While no existing public mmWave HPE dataset fully replicates the nonstationarity of real-world clinical deployment, the combination of controlled cluttered scenes (XRF55 multi-person), diverse motion primitives (mmRadPose), and cross-dataset generalization experiments (HuPR → mmRadPose, Appendix E) collectively provides a systematic stress-test of the proposed method.
>
> We acknowledge the gap between current benchmarks and real-world deployment as an important motivation for our planned future data collection, as noted in the Conclusion, and will incorporate this consolidated summary into the main text.
>
> **Q2: Clarification of the term "prompting".**
>
> We thank the reviewer for flagging this potential ambiguity. In this paper, "prompting" refers specifically to gated conditional cross-attention feature modulation: Doppler tokens, after reliability screening via learned confidence gates, are injected into spatial token reasoning through locality-constrained cross-attention. This mechanism is unrelated to prompt engineering in large language models. The term emphasizes the asymmetric, one-directional, conditional nature of the information flow — Doppler serves as a hint that guides spatial reasoning rather than a symmetric input channel. We will add a clarifying parenthetical at the first use of the term in the revised manuscript.
>
> We hope these clarifications fully address the reviewer's questions and are grateful again for the constructive and detailed feedback.

---

> > ### Author Rebuttal · Reviewer_RVDF · 2026-04-01
> >
> > The authors have fully addressed both points raised in the review. The dataset characteristics are now clearly described, including scale, diversity, and relevance to the motivating application scenarios, which improves interpretability of the experimental results. The clarification of the term “prompting” resolves the potential ambiguity. These were minor concerns and have been satisfactorily addressed.

---

### Decision · Program_Chairs · 2026-04-30

**Decision:**

Accept (regular)

**Comment:**

This paper studies the temporally stable mmWave human pose estimation. It has a clear technical framing and solid empirical support. A central concept explored by the paper is the use of Doppler as a reliability gated motion cue that conditions spatial reasoning instead of being fused symmetrically, and the results show consistent gains in both pose accuracy and temporal consistency across multiple datasets. The research's principal contribution is a lightweight prompting mechanism that improves stability without relying on tracking or post hoc smoothing, supported by careful ablations and useful plug in results. Most reviewer concerns were addressed well in the rebuttal, and while one reviewer remained somewhat cautious on a few details, the overall reviewer consensus is positive.